# Partner choice, confounding and trait convergence all contribute to phenotypic partner similarity

Jennifer Sjaarda[1,2,3] & Zoltán Kutalik ®[1,2,3] ✉

Partners are often similar in terms of their physical and behavioural traits, such as their education, political affiliation and height. However, it is currently unclear what exactly causes this similarity—partner choice, partner influence increasing similarity over time or confounding factors such as shared environment or indirect assortment. Here, we applied Mendelian randomization to the data of 51,664 couples in the UK Biobank and investigated partner similarity in 118 traits. We found evidence of partner choice for 64 traits, 40 of which had larger phenotypic correlation than causal effect. This suggests that confounders contribute to trait similarity, among which household income, overall health rating and education accounted for 29.8, 14.1 and 11.6% of correlations between partners, respectively. Finally, mediation analysis revealed that most causal associations between different traits in the two partners are indirect. In summary, our results show the mechanisms through which indirect assortment increases the observed partner similarity.

People in partnerships are more similar to one another than randomly sampled pairs. Partners tend to be similar with respect to traits such as various anthropometric measures (body mass index (BMI) and height), socioeconomic factors, behaviours (religious views[1] and social attitudes[2]), lifestyle (diet, smoking habits and hobbies) and even disease risk[3–10].

Several different causes can explain this phenotypic similarity. First, people actively look for partners who are similar to them[11,12], a phenomenon known as assortative mating (AM). Second, phenotypic similarity can reflect trait convergence during the partnership. In this case, traits become more similar over time because partners share a household or they influence each other's behaviour[13–15]. Third, partner similarity may be caused by confounders at the time of partner choice (or later), such as shared sociocultural environment and geographical barriers[16–18] (Fig. 1). Indirect assortment is a special case of the latter, where the confounder is the correlated trait which people use in direct partner selection[19].

In couples, one person's genome correlates with certain traits of their partner[20], and one study found evidence of direct genetic associations between the genome of an individual and the phenotypes of their partner. This finding suggests that partner heritability of a trait cannot be explained by between-trait correlation alone[21]. Overall, causes and consequences of phenotypic assortment remain unresolved.

The causes of partner similarity matter for the fields such as behavioural science, population genetics and public health. For instance, high phenotypic similarity could imply genetic similarity. In this case, otherwise independent gene variants would become correlated and would ultimately increase genotype homozygocity[22]. Partner similarity also affects the studies of genetic associations; it increases the estimates of heritability[23] and genetic correlation[24], even in ACE (additive, common environment and unique environment) models[25]. Moreover, partner similarity can introduce collider bias in within-spouse association models[26].

Similar to classical epidemiological studies where it is difficult to discern causal factors from confounders, mere phenotypic similarity within couples poses interpretational challenges. Mendelian randomization (MR) is a special case of instrumental variable (IV) analysis, whereby genetic markers are used as instruments to infer causal relationship. Because of the random allocation of genetic variants at birth,

[1]University Center for Primary Care and Public Health, Lausanne, Switzerland. [2]Department of Computational Biology, University of Lausanne, Lausanne, Switzerland. [3]Swiss Institute of Bioinformatics, Lausanne, Switzerland. ✉e-mail: zoltan.kutalik@unil.ch

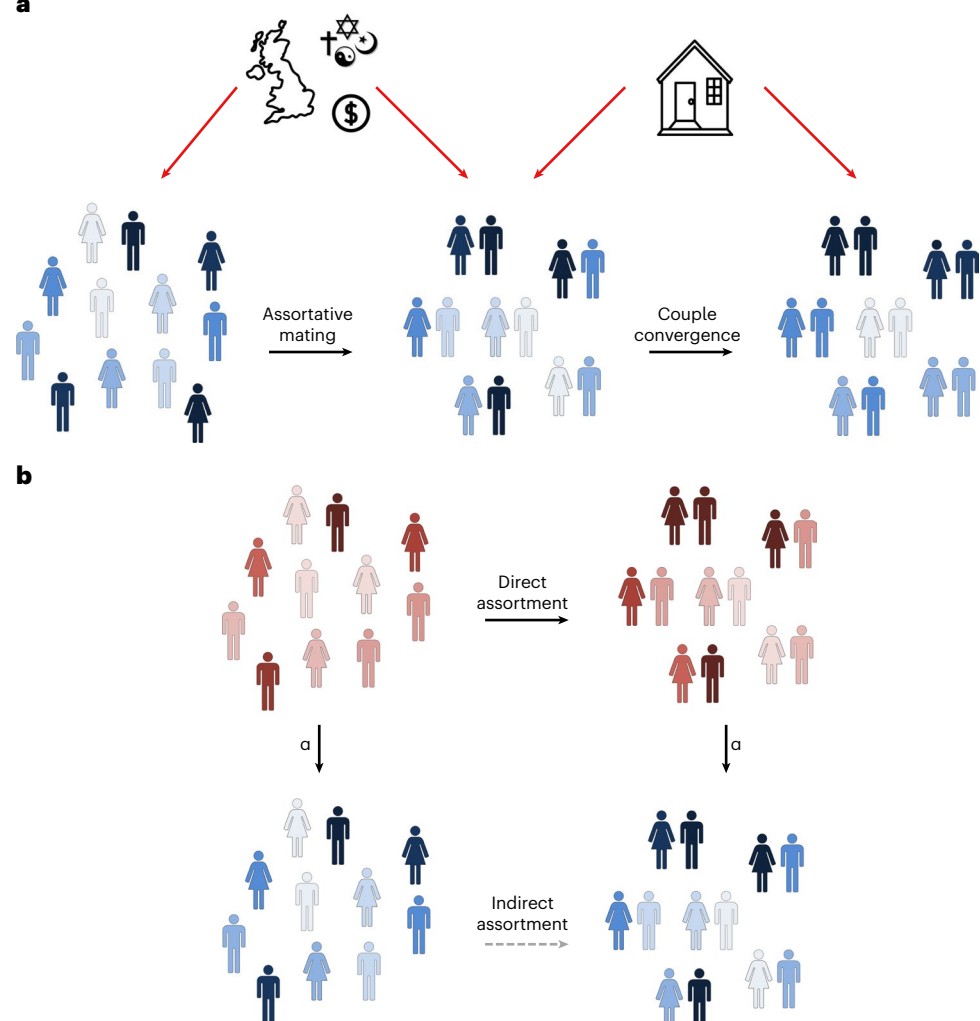

**Fig. 1 | Partner similarity framework. a**, Illustrates a trait (given by the colour blue) which shows increased similarity between partners, either directly (through mate choice) or due to confounding factors such as shared geography, cultural or religious status or socioeconomic measures. Subsequently, this trait may also undergo postmating convergence, which could be due to direct causal influence from one partner on the other (that is, through imitation or influence) or due to confounding factors such as shared environment. **b**, Illustrates a trait which shows increased similarity among couples (given by the blue trait); however, this assortment is only observed because of a causal effect (α) that exists between another trait (shown in red) acting on the blue trait. For example, if direct assortment occurs under a trait such as BMI (that is, couples intentionally select partners of similar BMI as themselves), phenotypic correlation will also be observed at all traits which have a causal effect on BMI, such as blood pressure, fasting glucose and so on.

MR can infer causality between an exposure and an outcome[27], avoiding reverse causality and confounding.

In this work, we used MR to study causality in couples where the exposure and outcome traits belong to different people (Fig. 2a). This concept is different from classical MR designs within individuals, such as the study of BMI causally affecting the risk for coronary artery disease[28]. The authors of ref. [29] used an approach similar to ours to study partner similarity in alcohol use. They showed that phenotypic correlation in couples does not increase with age and there was a difference between correlation and the direct causal effect[29].

Here, we applied MR to estimate the causal effects between partners for 118 traits. We studied direct effects on trait similarity, the impact of time couples have lived together and the role of confounders. Finally, we explored how cross-trait partner similarity emerges by dissecting them to direct and indirect parts.

## Results

This study analysed data from the UK Biobank (UKBB) cohort, a prospective population-based study with over 500,000 adult participants.

Among them, 51,664 couples were identified and selected according to a procedure described in the Methods and Supplementary Fig. 1. Starting from 1,278 available phenotypes, we selected those with between-partner (Pearson) correlation larger than 0.1 and having at least five valid instruments. Pearson and Spearman correlations led to very consistent estimates (Supplementary Fig. 2). These were further filtered (Methods and Supplementary Fig. 3) to yield 118 traits to analyse.

### Effect of sex, age and time spent together

Among the 118 phenotypes tested, we identified 64 significant ($P < 0.05/66$) causal effects in partners after adjusting for the effective number of tests (66) (Supplementary Table 1). We also examined the Cochran's heterogeneity $Q$ statistic to identify traits with high heterogeneity and found no evidence of heterogeneity in the MR estimates (all $P > 0.05/66$). We assessed the 64 significant results for sex differences but did not identify any after adjusting for the effective number of tests among the remaining traits based on their pairwise correlation matrix ($P < 0.05/29$).

To identify if partner traits converge over time, we explored the impact of age and time spent together (proxied by the amount

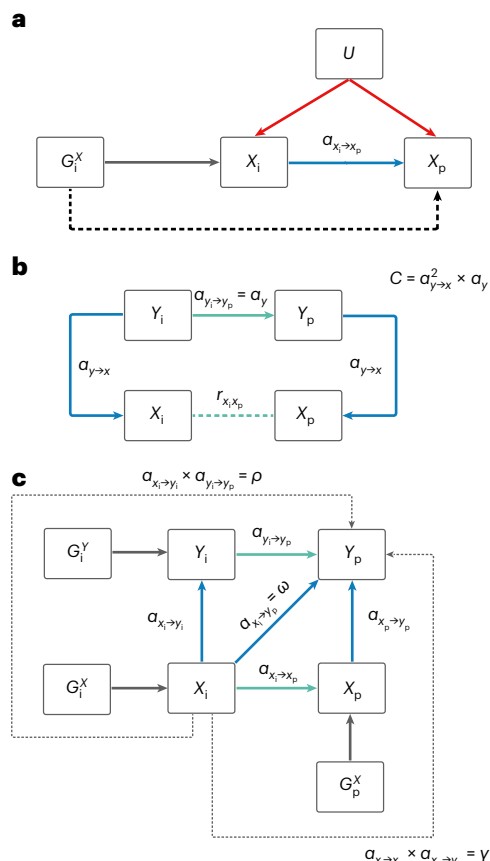

**Fig. 2 | MR schematic within couples. a**, Illustrates the causal effect among couples with a single trait ($\alpha_{x_i \to x_p}$), where $G$ represents genetic variant(s), $X$ represents a single trait (in an index individual ($X_i$) and a partner ($X_p$)) and $U$ represents confounding factors that are not associated with genetic variance owing to the random distribution of alleles at conception. Throughout the paper subscript i and p refer to the index and the partner, respectively. **b**, Directed acyclic graph illustrates the impact a confounder (trait $Y$) could have on the phenotypic correlation between partners for a given trait $X$ ($r_{x_i x_p}$). Correlation due to confounding can be calculated as $C = \alpha_{y \to x}^2 \times \alpha_y$. **c**, Represents the expanded causal network involving two traits and the various estimated causal paths from trait $X$ of an index case ($X_i$) to a phenotype $Y$ in the partner ($Y_p$) given by $\omega$, $\gamma$ and $\rho$. Cross-trait causal effects from $X_i$ to $Y_p$ ($\omega$) can be summarized by three possible (non-independent) scenarios: (1) $X_i$ could exert a causal effect on $X_p$, followed by $X_p$ having a causal effect on $Y_p$ in the partner alone ($\gamma$); (2) the reverse could occur whereby $X_i$ has a causal effect on $Y_i$ in the index alone, followed by a causal effect of $Y_i$ case on $Y_p$ ($\rho$); or (3) there could be other mechanisms, either acting directly or through other unmeasured or unconsidered variables. To quantify $\rho$, we first estimated the causal effect of $Y_i$ on $Y_p$ in MVMR (not illustrated) to exclude any residual effect of $X$ on phenotype $Y$ from index to partner. These three scenarios could also act in some combination. Therefore, the $\omega$ estimate would capture the paths of $\gamma$, $\rho$ and other mechanisms combined. In both **a** and **c**, cross-partner causal effects are given by blue arrows and same-person causal effects are given by green arrows.

of time at the same address) among the 64 significant traits in both men and women separately and both sexes combined. Using linear regression of MR estimates versus the median of the five age and/or time spent together bins, we detected no significant results in the sex-combined results after adjustment for the number of effective tests ($P < 0.05/66$). We also examined the Pearson phenotypic correlation within the different bins and assessed for the presence of a trend using linear models (phenotypic correlation versus median bin) for all 118 phenotypes. Two traits showed a significant ($P < 0.05/66$) trend across the bins according to time spent together, namely body

fat percentage (slope $= -0.0018$, $P = 1.96 \times 10^{-5}$) and forced expiratory volume in 1 second (slope $= -0.0043$, $P = 2.98 \times 10^{-4}$). In both cases, the correlation decreased as time spent together increased. We found two other traits that showed a significant trend across the bins by median age, namely previous smoking status (slope $= 0.0011$, $P = 6.9 \times 10^{-4}$) and aspirin use (slope $= 0.0015$, $P = 1.8 \times 10^{-4}$). In this case, for both phenotypes, the slope increased as median age increased (Fig. 3 and Supplementary Table 1). Using Spearman correlation yielded consistent results (all $P < 0.05/66$).

### Causal effects versus raw phenotypic correlations

To better understand the nature of phenotypic assortment, we assessed whether there were any discrepancies between the causal effect estimates within couples ($\hat{\alpha}$) and observational correlations ($\hat{r}$). Using MR, the causal effects between partners (within couples) were estimated for 118 phenotypes. These traits were selected based on their elevated correlation between partners and sufficient (more than five) valid IVs, making them suitable for MR analysis. Using a two-tailed $Z$ test to gauge the statistical significance of the difference between the estimates (with test $P$-value denoted by $P_{diff}$), we compared (standardized) causal MR effects to the raw phenotypic correlation among couples to identify any traits where the correlation was different than the MR estimate. After adjusting for the effective number of tested traits ($P < 0.05/66$), we identified 43 traits which showed different phenotypic correlation compared to MR estimate (Fig. 4a and Supplementary Table 1). Of these, three had a larger MR estimate compared to correlation (time spent watching television, comparative height size at age 10 and overall health rating), while the remaining 40 traits had a larger (absolute) correlation compared to MR estimate. These included place of birth, north coordinate (NC; $\hat{r} = 0.58$ versus $\hat{\alpha} = 0.33$, $P_{diff} = 2.47 \times 10^{-18}$), systolic blood pressure ($\hat{r} = 0.16$ versus $\hat{\alpha} = 0.05$, $P_{diff} = 4.89 \times 10^{-9}$), height ($\hat{r} = 0.25$ versus $\hat{\alpha} = 0.21$, $P_{diff} = 6.63 \times 10^{-6}$), forced vital capacity ($\hat{r} = 0.25$ versus $\hat{\alpha} = 0.13$, $P_{diff} = 5.62 \times 10^{-16}$), basal metabolic rate ($\hat{r} = 0.21$ versus $\hat{\alpha} = 0.16$, $P_{diff} = 9.78 \times 10^{-7}$) and basophil count ($\hat{r} = 0.47$ versus $\hat{\alpha} = 0$, $P_{diff} = 1.76 \times 10^{-38}$) (Fig. 4a).

Significant differences could be indicative of the presence of confounders (either negative or positive) driving the observed phenotypic correlation. Thus, for traits where couple correlation was significantly different than MR causal estimates, we sought to identify potential confounders which may, in part, explain the discrepant estimates. For the three traits where correlation was less than the MR estimate, we searched for negative confounders (that is, negative $\alpha_{y_i \to y_p}$) but did not identify any.

Conversely, for traits where the correlation was greater than the MR estimate, we searched for positive confounders and found many potential positive confounders. Namely, the mean number of potential confounders from our set of 117 candidates was 22.56, with a maximum of 39; there was only one trait for which we did not identify any potential confounders (Supplementary Table 2). For instance, for systolic blood pressure, we identified 29 (correlated) potential confounders, which may explain the larger phenotypic correlation ($\hat{r} = 0.16$) as compared to MR effect ($\hat{\alpha} = 0.05$). These potential confounders included physical activity, BMI, lung fitness measures and overall health rating. For weight ($\hat{r} = 0.23$ versus $\hat{\alpha} = 0.19$), we found 30 potential confounders, including anthropometric traits (such as leg, trunk and arm fat mass) and various behavioural traits which are reflective of exercise patterns, such as time spent watching television, walking pace and phone use, among many others (Supplementary Table 2). Many of the 40 traits with larger phenotypic correlation compared to MR estimates included blood cell counts and/or percentages (such as white blood cell (leucocyte) count, neutrophil count, monocyte count and percentage and reticulocyte percentage and count). The potential confounders for these traits were highly overlapping, including physical activity level, anthropometric traits, smoking and health rating (Supplementary Table 2). Other notable confounders included measures of physical

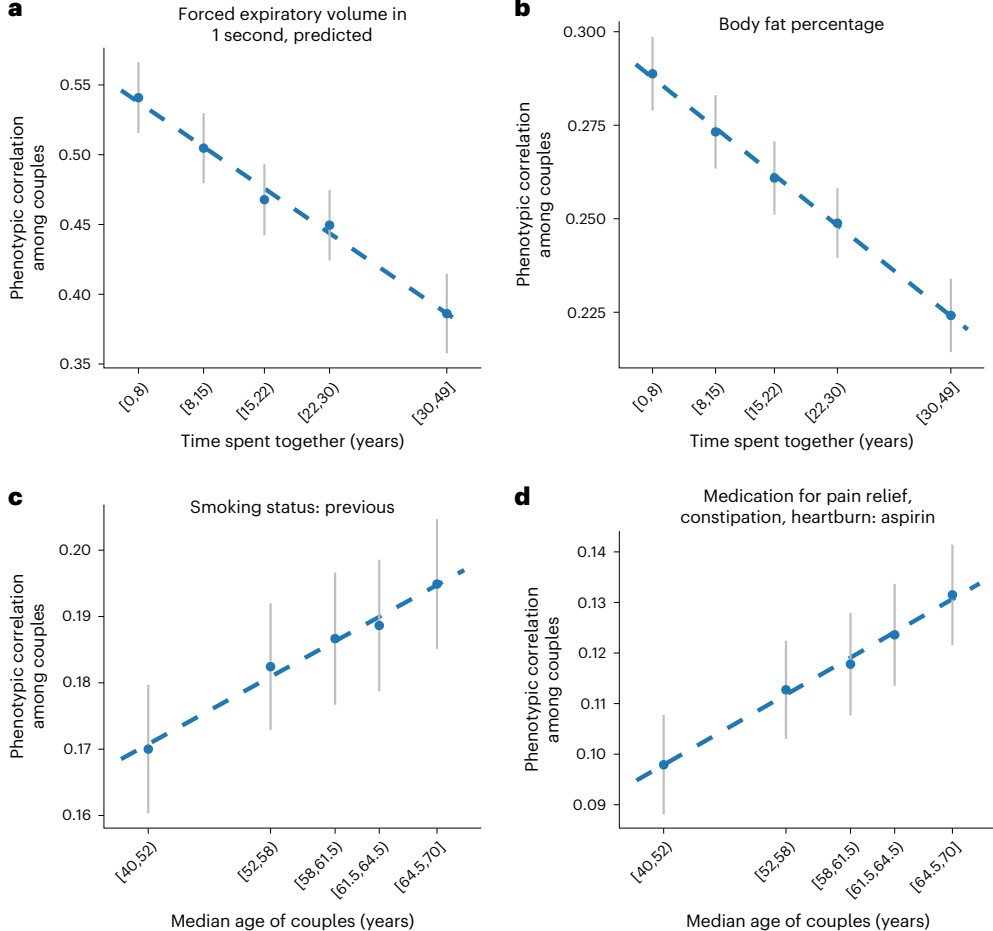

**Fig. 3 | Phenotypic correlation for selected traits by time spent together and age of couples. a–d,** Scatterplots show the phenotypic correlation for four selected traits among couples within different bins. Couples were binned by time spent together (proxied by the time lived at the same household) (**a,b**) and median age (**c,d**). The four panels show correlations for four different traits: forced expiratory volume (**a**), body fat percentage (**b**), previous smoker (**c**) and aspirin use (**d**).

activity for forced vital capacity; smoking status and fitness measures for basal metabolic rate; and measures of body size for hand grip strength.

Finally, for each confounder we calculated the correlation due to confounding (*C*) as described above (Methods and Figs. 1b and 2b) and summed up these *C* values for all uncorrelated confounders. We then, for each trait, compared the difference in estimates (that is, $\hat{r} - \hat{a}$) to the estimated value $C_{sum}$ (Fig. 4b). One can observe some traits (for example, systolic blood pressure) where the difference between partner correlation and causal effects can be well explained by the tested confounders, but for the majority of the traits, the observed confounders are not sufficient to account for the discrepancy (for example, basophil count has strong positive confounders missing).

## Major confounders of trait correlations

Next, we assessed the impact of potential confounders on trait correlation in couples by calculating the ratio of correlation due to confounding over the raw phenotypic correlation among couples averaged across all traits tested (Supplementary Table 3). While geographical location (using place of birth north/east coordinates) was found to have a negligible impact on phenotypic correlations (mean confounding ratio 1%), household income (mean confounding ratio 29.8%), age completed full-time education (mean confounding ratio 11.6%) and physical activity levels (measured using the variable 'leisure/social activities: sport club or gym'; mean confounding ratio 17.1%) had an

important confounding impact on raw phenotypic correlation among couples (Fig. 5).

## Cross-trait assortment

We sought to identify the mechanisms underlying partner similarity by comparing three estimated paths from a phenotype in the index case ($X_i$) to another phenotype in its partner ($Y_p$) as illustrated in Fig. 2c. The total causal effect between $X_i$ and $Y_p$ (denoted by $\omega$) can be split up into three components: (1) AM through $X$ (that is, $X_i \rightarrow X_p$) followed by a causal effect between $X$ and $Y$ in the partner (that is, $X_p \rightarrow Y_p$), their product being denoted by $\gamma$; (2) causal effect between $X$ and $Y$ in the index individual (that is, $X_i \rightarrow Y_i$), followed by AM through $Y$ (that is, $Y_i \rightarrow Y_p$), their product being denoted by $\rho$; and (3) any remaining effect of $X_i$ on $Y_p$.

We computed within-couple cross-trait causal effect estimates $X_i \rightarrow X_p$ (that is, $\hat{\omega}$) for all combinations of trait pairs ($X, Y$). Of these, we identified 1,327 significant MR effects ($p_{\hat{\omega}} < 0.05/[66^2]$) among couples, which were reduced to 1,088 pairs after removing pairs with phenotypic correlation >0.8 (a summary of a set of pruned traits can be found in Supplementary Table 4). Several relationships were almost completely dominated by $\rho$ (AM through the outcome) and others dominated by $\gamma$ (AM through the exposure). Specifically, we found 326 relationships which were significantly different between $\rho$ and $\gamma$, of which 89 (27.3%) showed larger effects through $\rho$ and the other 237 (72.7%) showed larger effects through $\gamma$. For instance, we found causal relationships between

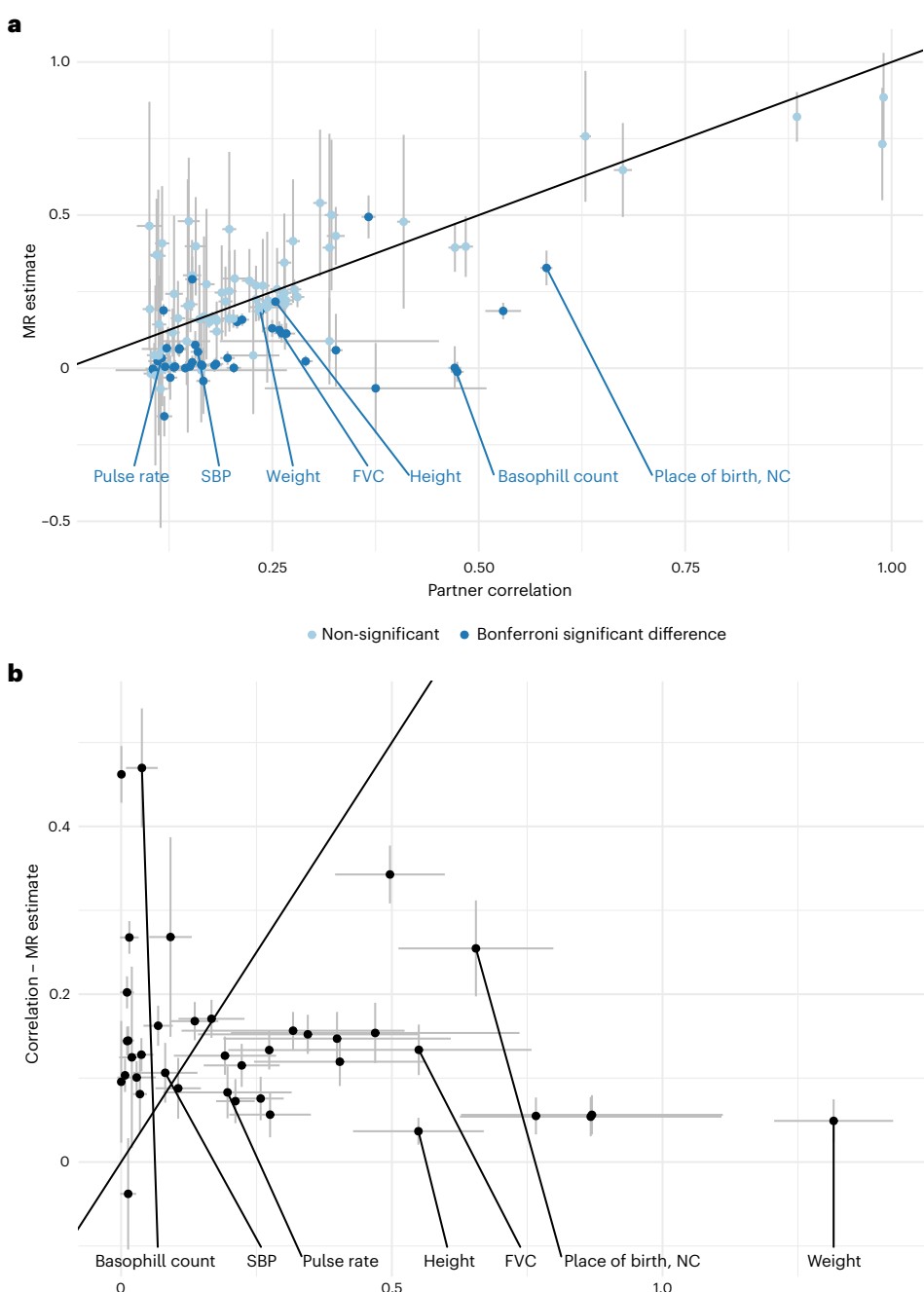

**Fig. 4 | Phenotypic correlation in couples versus causal effects and evidence of confounder traits impacting the discrepant estimates. a**, Scatterplot shows the within-couple standardized MR estimates ($\alpha_{x_i \to x_p}$) versus the phenotypic correlation among couples ($r_{x_i x_p}$). The centre of the confidence interval (CI) is the estimate for the corresponding parameter and error bars represent 95% CIs. A two-tailed $Z$ test was used to test for a significant difference between the estimates. After adjusting for the number of effective tests ($P < 0.05/66$), 43 significant differences were identified (shown in dark blue), where 3 traits showed larger MR estimates compared to correlation and 40 traits showed larger correlation compared to MR estimates. The identity line is shown in black. Labelled pairs are discussed in the main text. **b**, Scatterplot shows the difference in phenotypic correlation and MR estimate versus the $C_{sum}$ value (estimating the correlation induced by measured (uncorrelated) confounders) for each trait where the phenotypic correlation was greater than the MR estimate (number of traits = 39); error bars represent 95% CIs. The identity line is shown in black. FVC, forced vital capacity; NC, north coordinate; SBP, systolic blood pressure.

partners for leg fat percentage on the time spent watching television and BMI on overall health rating, all dominated by $\rho$.

On the other hand, we found some causal relationships between partners which were primarily dominated by $\gamma$ (AM through the exposure), including comparative height at age 10 (that is 'When you were 10 years old, compared to average would you describe yourself as shorter, taller, average'), forced vital capacity and standing height on hand grip strength. Finally, we found other pairs where neither $\rho$ nor $\gamma$ captured the relationship (that is, $\hat{\omega}$ was significantly larger than both estimates), including BMI effect on partner's systolic and/or diastolic blood pressure.

Finally, we estimated the contribution of the first two components ($\hat{\gamma}$ and $\hat{\rho}$) contributing to these significant cross-trait effects and compared their contribution to the total effect using standard linear

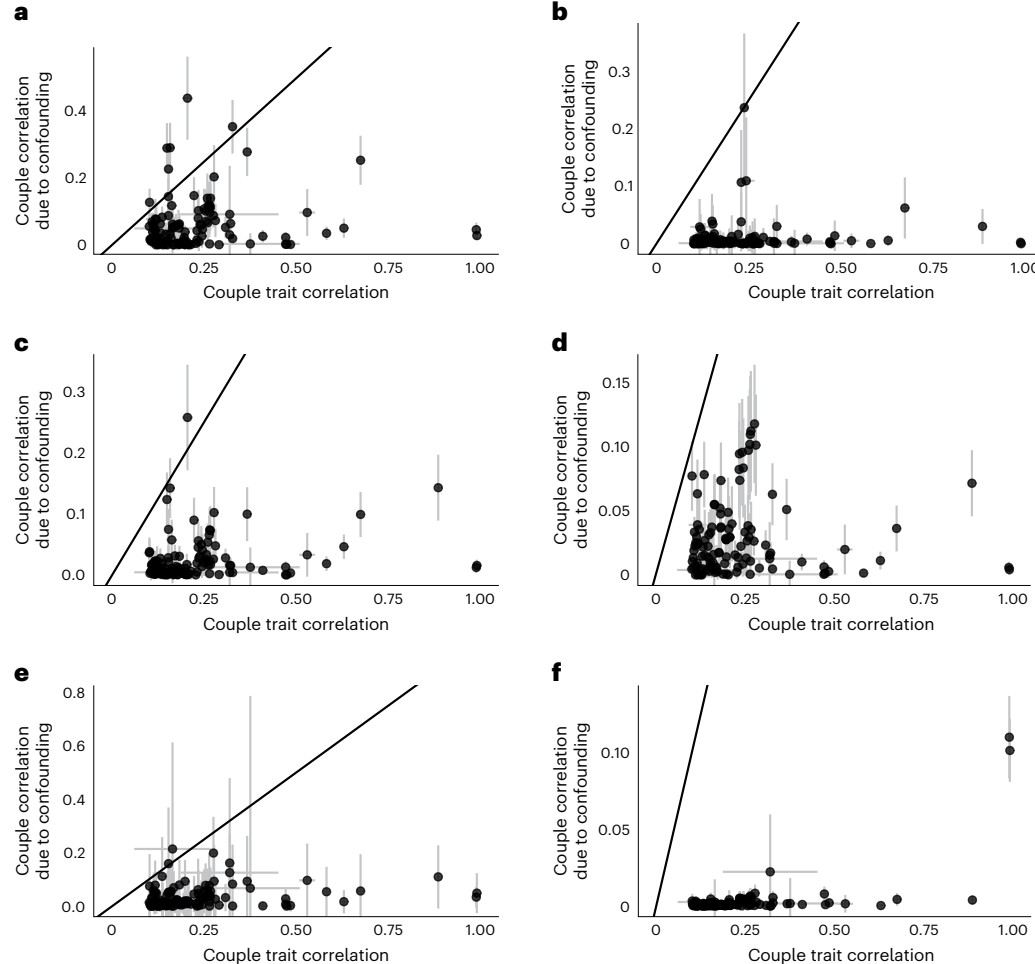

**Fig. 5 | Global confounding impact of select traits on couple phenotypic correlation. a–f**, Scatterplots of couple correlation due to confounding versus the phenotypic trait correlation among couples for selected potential confounder traits ($Z$). The couple correlation due to confounding for each trait $X$ was calculated for each confounder $Y$ as $C = \alpha_{y \to x}^2 \times \alpha_{y_i \to y_p}$. In the case of birthplace coordinates, $C$ values were summed across the two (independent) north and east coordinates. The centre of the CI is the estimator value and the error bars represent the 95% CI. These confidence intervals for the correlations shown on the $x$ axis are based on the number of couples shown in the 'n_pairs'

column of Supplementary Table 2. The CI for confounder-induced correlation was computed as 1.96 times the s.e. of the estimator $\hat{\alpha}_{y \to x}^2 \times \hat{\alpha}_{y_i \to y_p}$, the computation of which is described in the Methods. For each trait in the pipeline, we tested how the contribution of six confounder traits (average total household income before tax (**a**), current tobacco smoking (**b**), age completed full-time education (**c**), overall health rating (**d**), sports club or gym user (**e**) and place of birth coordinates (**f**)) could impact the phenotypic couple correlation. The identity line is shown in black.

regression (Fig. 6 and Supplementary Table 5). Paired $t$ test comparing $\hat{\gamma}$ and $\hat{\rho}$ effect estimates revealed that $\hat{\gamma}$ (AM through $X$) is stronger ($P = 1.1 \times 10^{-5}$) in general compared to $\hat{\rho}$ (AM through $Y$). When we summed up the effects of $\hat{\gamma}$ and $\hat{\rho}$, we found that the sum was significantly larger than $\hat{\omega}$. However, these two effects seemed to be correlated, carrying potentially shared signals. Hence, we first residualized $\hat{\rho}$ for the effects of $\hat{\gamma}$ ($\widehat{\rho_{resid}}$) to ensure independence between the two estimates and then added $\widehat{\rho_{resid}}$ to $\hat{\gamma}$ ($\widehat{\rho_{resid}} + \hat{\gamma}$). We found no significant difference between $\hat{\omega}$ and the sum of $\widehat{\rho_{resid}} + \hat{\gamma}$ in this analysis and with data points in general falling near the identity line, suggesting that $\hat{\omega}$ was capturing the paths given by $\hat{\gamma}$ and $\hat{\rho}$. Indeed, linear regression results revealed that 76% of the total effect ($\hat{\omega}$) can be explained by the two paths ($\widehat{\rho_{resid}} + \hat{\gamma}$) and that the $\widehat{\rho_{resid}} + \hat{\gamma}$ is on average very close to the total effect.

### The extent of bias in the MR estimates
In the Methods, we formulated a general model that accommodates parental effects and direct, indirect assortment (Fig. 7). We then, assuming this model, derived the analytical formula for the bias of

cross-sample MR estimates, which shows its exact dependence on each model parameter. Here, we explored the extent of the bias under realistic ranges of model parameters and visualized it in Fig. 8. First, we observed (Fig. 8a) that $r_G$ (the correlation of the genotype between partners) has little impact on the bias (since it is limited by the heritability and the genetic correlation) compared to direct environmental assortment ($r_E$), and they contribute additively. A similar relationship can be observed (Fig. 8b) for the parental genetic- ($s_G$) and parental trait effects ($s_X$) on the offspring's environment; the impact of the former is dwarfed by the latter for the same reason. Unsurprisingly, the largest bias emerges when both the $r_E$ and $s_X$ are the largest (Fig. 8c). This can be complemented by $r_E$ combined with parental genetic effects on the offspring's environment ($s_G$) (Fig. 8d).

Overall, when direct environmental or genetic assortment is moderate, parental effects lead to negligible bias in the causal effect estimate between $X_O^i$ and $X_O^p$. This is why we focused our real data analysis efforts on identifying such $X$-associated (confounding) factors for which direct assortment may occur, so that $r_E$ (and hence, the MR bias) could be limited.

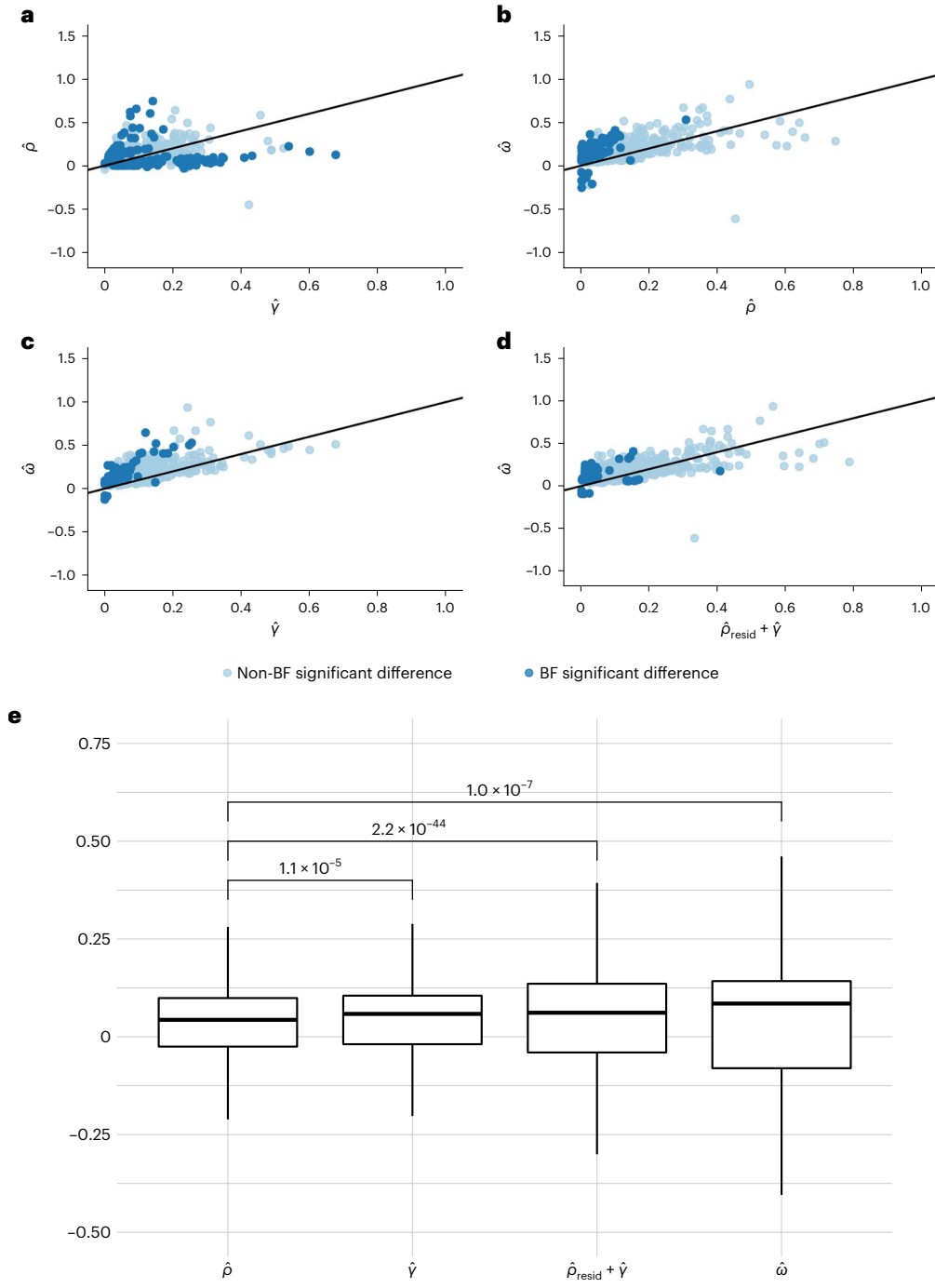

**Fig. 6 | Comparison of causal paths between two traits within couples.**
**a**–**d**, We estimated various causal effect paths ($\rho$, $\gamma$ and $\omega$, see Fig. 2c) from a phenotype of the index case ($X_i$) to another phenotype of its partner ($Y_p$) for the 1,088 trait pairs with significant MR effects among couples ($p_{\hat{\omega}} < 0.05/[66^2]$) and trait pair correlation <0.8. Panel (**a**) provides a scatter plot for $\hat{\rho}$ against $\hat{\gamma}$; panel (**b**) for $\hat{\omega}$ against $\hat{\rho}$; panel (**c**) for $\hat{\omega}$ against $\hat{\gamma}$ and panel (**d**) for $\hat{\omega}$ against $\widehat{\rho_{resid}} + \hat{\gamma}$. The solid black line represents the linear regression fit. Dark blue dots indicate trait pairs with significant (after Bonferroni, BF, correction) difference between the respective parameters shown in the scatter plot, while light blue one mark the remaining traits. To calculate $\widehat{\rho_{resid}} + \hat{\gamma}$, we residualized $\hat{\rho}$ for the effects of $\hat{\gamma}$ ($\widehat{\rho_{resid}}$) to ensure complete independence between the estimates and then added $\widehat{\rho_{resid}}$ to $\hat{\gamma}$ ($\widehat{\rho_{resid}} + \hat{\gamma}$). **e**, A box plot comparing the coefficients of the estimates among the trait pairs after removing 19 trait pairs where the sign did not match between any combination of the four coefficients. In the box plots, the lower and upper hinges correspond to the first and third quartiles and the middle bar corresponds to the median; the upper whisker is the largest point smaller than 1.5 times the interquartile range above the third quartile; the lower whisker is defined analogously. We used a two-sided paired $t$ test to compare the presented estimates ($\hat{\rho}$, $\hat{\gamma}$, $\widehat{\rho_{resid}} + \hat{\gamma}$ and $\hat{\omega}$).

## Discussion

In this article, we studied causal relationships behind trait similarity within couples by applying MR to the UKBB data. We analysed 118 traits, representing a wide range of anthropometric-, behavioural- and disease-related traits. Among the 118 phenotypes tested, we found widespread evidence of causal effects among partners. In particular, we identified 64 same-trait causal effects within partners (of 118 traits) and no evidence of heterogeneity among same-trait couple MR estimates ($\alpha_{x_i \rightarrow x_p}$). This suggests that associations between a person's genotype and their partner's phenotype are primarily acting indirectly

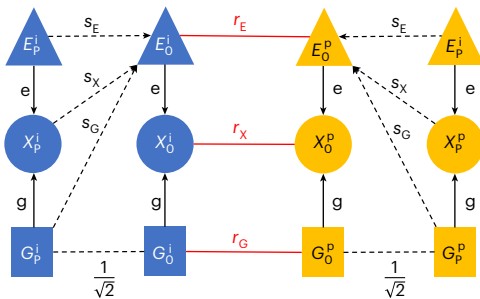

**Fig. 7 | Modelling the impact of parental effect and AM on cross-sample MR.** The diagram represents the underlying joint model of parental effects and assortment. A focal trait (X) has genetic (G) and enviromental (E) components, with effect size $g$ and $e$, respectively. Their subscripts can be O, F or M, referring to the offspring, the father or the mother, respectively. The superscripts can be either $i$ or $p$, indicating the index individual or its partner. We allowed parental genetics, parental environment and the parental trait each to influence the offspring's environment, with corresponding direct effect strengths $s_G$, $s_E$ and $s_X$, respectively. Finally, couples are formed under direct assortments acting on G, E and X, leading to correlations $r_G$, $r_E$ and $r_X$, respectively.

through the cross-partner causal relationship between the trait(s) associated with the genotype, rather than the presence of a direct effect for index genotype to the partner's phenotype.

Our results suggest that fitness and anthropometric measures are important initially (at the time of mate choice), but their correlation decreases with time; the longer partners stay together, the less important it gets to remain similar. On the other hand, we found that couples become more similar with respect to smoking cessation and aspirin use as their age increases. As age and time spent together are highly correlated, it is difficult to distinguish whether this is an effect of convergence or of age-dependent partner choice. We did not find any significant trends of causal MR effects on time spent together or by age. While this could be because of limitations such as statistical power, this is consistent with previous reports which suggest that initial partner choice is more important than convergence[9,30–32].

Deriving an analytical formula for the bias in the cross-partner MR estimation allowed a detailed analysis of model parameters that contribute most to such potential bias. Our analysis revealed that the largest contribution to the bias is direct environmental assortment combined with strong parental effects. This conclusion prompted us to explore potential confounders for each examined trait.

When investigating the impact of common confounders on our entire panel of phenotypes (that is, fixing a confounder and assessing its widespread impact on all single-trait AM), we found that household income, age completed education and participant of a sport club or gym are important confounders, explaining on average 29.8, 11.6 and 17.1% of the phenotypic couple correlations among traits tested, respectively. These results also suggest that phenotypic correlations in couples are significantly confounded and point to a relatively few key traits which are driving observed partner similarity. These confounder traits are strongly intertwined and hence correlated, therefore elucidating that the key driver is not feasible with the data at hand. Overall, we noticed that the tested confounders are not sufficient to account for the gap between couple correlations and causal effects or the latter being incorrectly estimated. Of note, phenotypic correlations in couples are impacted differently by measurement noise than causal effect estimates. While the former estimates are attenuated by a factor of $\frac{\text{Var}(y)}{\text{Var}(y)+s^2}$ in case the true phenotype $y$ is measured with a noise with variance $s^2$, the causal effect estimates do not change noticeably because the exposure and outcome effects are equally diluted. This could lead to an underestimation of confounding effects in our results and may explain why for three traits we observed larger causal effect than correlation.

Our findings investigating cross-trait assortment suggest that causal effects from $X_i$ to $Y_p$ are primarily driven by AM through $X$ (that is, $X_i \rightarrow X_p$) followed by a causal effect within the partner from $X$ to $Y$ (that is, $X_p \rightarrow Y_p$). In contrast, a less likely path would be the inverse, whereby the presence of a causal effect from $X$ to $Y$ in an index case is then followed by $Y$ being passed directly from index to partner. These results were expected, as it is more reasonable for couples to influence each other at the exposure level rather than the outcome level, especially since often outcome traits (such as diseases) appear much later than mate choice.

We found 1,088 significant cross-trait causal effects within couples ($\omega$), which can be summarized by three categories: (1) driven by assortment on the exposure ($\omega = \gamma$), (2) driven by assortment on the outcome ($\omega = \rho$) and (3) not explained by either (that is, $\omega$ being greater than both $\rho$ and $\gamma$). Of note, there were fewer cases in category 3, where the causal effect from $X_i$ to $Y_p$ was not captured by $\gamma$ or $\rho$, suggestive of either a direct effect $X_i$ to $Y_p$ or indirect effects through variables we have not explored. An example from the first category involves a positive causal effect of time spent watching television on BMI driven by the fact that partners causally influence each other with respect to time spent watching television, which, in turn, has an impact on BMI at the individual level. On the other hand, an example of the second category includes a positive causal relationship from height to education, with a stronger path through $\rho$, whereby height (a proxy for 'dynastic' wealth) increases educational attainment (found previously[33]) within a single individual and AM subsequently occurs via education level. Finally, as an example for category 3, we found a negative causal effect of never having smoked on leucocyte count within partners, such that leucocyte count was higher among individuals with partners who smoked. While we also identified a significant effect through $\gamma$ (AM through smoking), the effect was much stronger through $\omega$. These findings suggest that there could be a direct effect from index partner by way of secondhand smoke. These results are consistent with previous work showing higher white blood cell count in smokers[34], which might already be achieved by secondhand smoking.

This study has limitations which should be acknowledged. First, to increase statistical power and robustness, we focused on traits available in the UKBB with significant correlation among couples and more than five valid IVs. As a result, anthropometric traits constituted a larger proportion of our traits under study and represent a large percentage of our significant findings. Other phenotypes, such as behavioural and lifestyle traits, were included but had less statistical power due to lower couple correlation and fewer IVs.

Second, using our data, we could not find strong evidence for couple convergence over time. This can be due to these effects being weak or the available data being suboptimal. Indeed, we used data on both age and time together (proxied by time at the same address) to answer this question, but these are poor proxies of the relevant traits. To properly disentangle the relationship between AM and convergence, additional longitudinal data including phenotypes at the time of partner selection would be necessary. A complementary approach could be to contrast genetic and phenotypic correlations; however, it is hard to tell whether differences reflect postmating effects or different (genetic and environmental) correlation to traits under primary assortment[20].

Third, we have not explored indirect causal paths from $X_i$ to $X_p$ through another variable ($Z$) measured in either individual. Given the limited evidence of direct cross-trait, cross-partner effects, the most likely such path would be $X_i \rightarrow Z_i \rightarrow Z_p \rightarrow X_p$, leading to an indirect effect of $\alpha_{X \rightarrow Z} \alpha_{Z \rightarrow X} \alpha_{Z_i \rightarrow Z_p}$. This implies a bidirectional causal effect between $X$ and $Z$ (within the same individual), but their product is expected to negligible[35]. Also, while assortment/convergence through the exposure ($\gamma$) and the outcome ($\rho$) represents independent paths from $X_i$ to $Y_p$, our results suggest that the computed effects using MR estimates are not perfectly independent. This could potentially be because of the overlap in genetic instruments or bidirectional causal effect between them or because both estimates depend on the causal

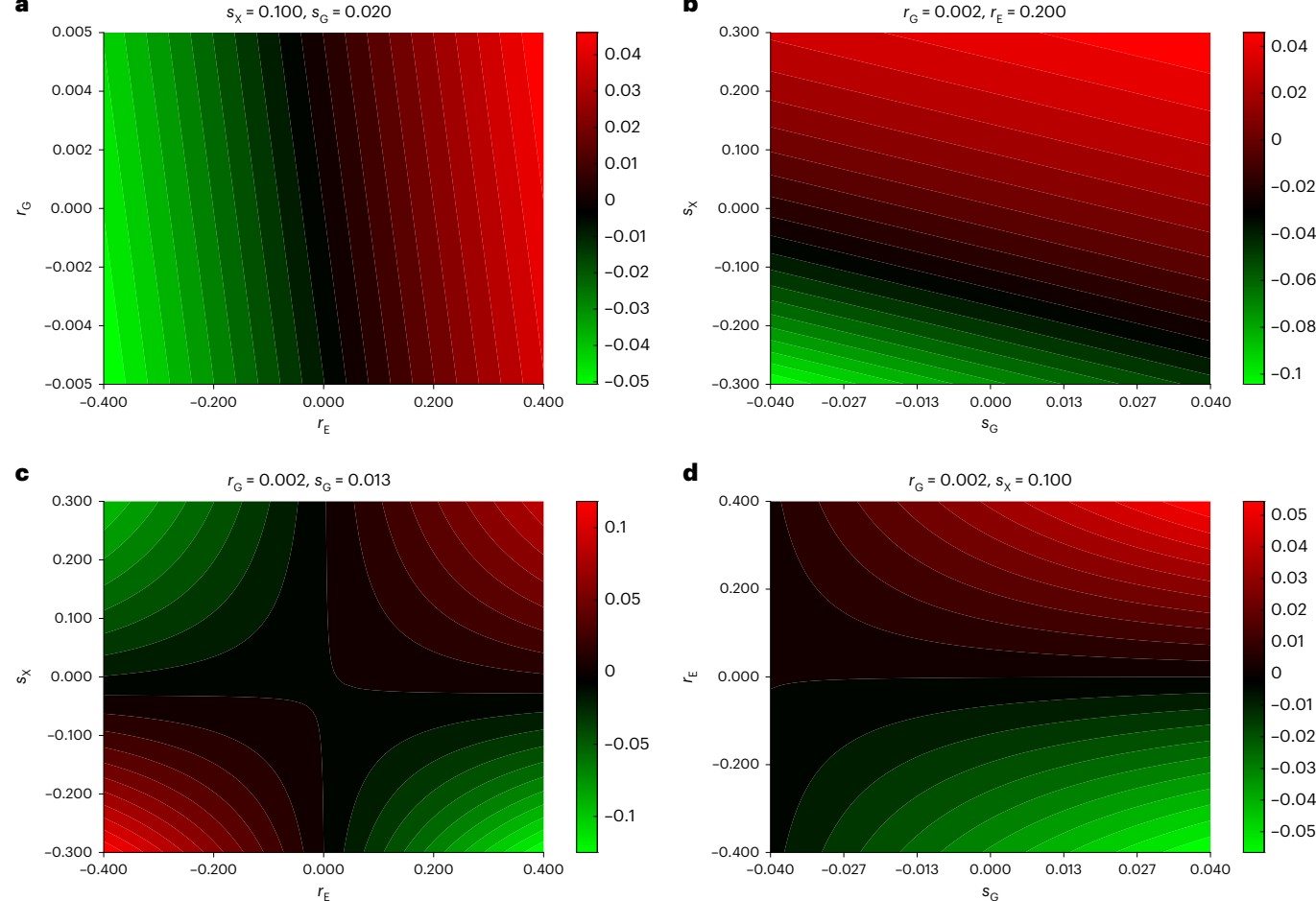

**Fig. 8 | The impact of various model parameters on MR bias.** We plotted the bias of the cross-sample MR estimates as a function of the proposed model parameters. Parameter $r_G$ refers to direct genetic assortment, $r_E$ to direct environmental assortment, $s_G$ to direct parental genetic effect and $s_X$ to direct parental trait effect. The different panels show the extent of bias when different pairs of parameters were covaried: $r_G$, $r_E$ (**a**), $s_G$, $s_X$ (**b**), $r_E$, $s_X$ (**c**) and $s_G$, $r_E$ (**d**).

effect from $X$ to $Y$. To the best of our ability, we tried to mitigate this bias by (1) using a multivariable Mendelian randomization (MVMR) approach to remove effects of $X$ on $Y$ in the calculation of $\rho$ and (2) first residualizing $\gamma$ for effects of $\rho$ to ensure independence before summation of the effects. Finally, we were limited to the available traits and white British samples in the UKBB. AM and partner selection are often population specific[36]. Therefore, our findings may not generalize to other populations, and more diverse biobanks are needed to systematically explore the heterogeneity in assortative behaviour.

In summary, we have surveyed 118 complex traits with significant couple correlation in the UKBB and explored the major contributors to the observed couple similarity: partner selection, couple convergence and confounding. We found that cross-trait assortment can largely be explained by single-trait assortments between either trait and substantial causal effects between these traits. Our findings provide insights into possible mechanisms underlying observed partner similarity patterns at an unprecedented scale and resolution.

## Methods
All analyses were run using the R software (v.3.6.3).

### Sample selection and couple definition
This study used the UKBB cohort, a prospective population-based study with over 500,000 adult participants. UKBB has approval from the North West Multi-Centre Research Ethics Committee as a Research

Tissue Bank approval. This approval means that researchers do not require separate ethical clearance and can operate under the Research Tissue Bank approval. UKBB also possesses a Human Tissue Authority licence, so a separate Human Tissue Authority licence is not required by researchers who receive samples from the resource.

Couples were identified and selected according to the following procedure. The initial UKBB sample comprised 502,616 individuals. First, participants were filtered to only genotyped, white, unrelated individuals according to the genetic quality control (QC) file. Redacted samples and participants who removed consent were also excluded. After filtering, 337,138 participants remained. Within this sample, we retained individuals coming from households with exactly two unrelated, opposite sex individuals, leaving 108,898 participants. Finally, using the data at data field 6,141, 'How are people in household related to participant' pairs were filtered to only include couples who had both responded 'Husband, wife, or partner', leaving 103,328 participants or 51,664 couples for downstream analyses (Supplementary Fig. 1). Note that some studies have used more stringent criteria for couple definition. For example, Yengo et al.[22] additionally requested couples to have been recruited in the same centre, living in the same location, for the same amount of time, living with the same number of people with the same household income, same Townsend deprivation, same number of smokers and so on. Howe et al.[29] used a very similar definition to ours (resulting in 10% fewer couples than us); however, they restricted their analyses to only couples who reported to live at the same address for

the same amount of time. We believe that discrepancies in self-reported data can occur frequently (due to misunderstanding, misreporting) and hence, decided to use a more liberal couple definition to increase sample size at the cost of minor misclassification.

## MR

MR uses genetic variants as IVs to assess the presence of a causal relationship. The random distribution of genetic variants at birth reduces the possibility of confounding or reverse causation as explanations for the link between the exposure and outcome in the same way that the random allocation of a therapy in a randomized controlled trial minimizes this risk. MR relies on three core assumptions for the genetic variants. First, IVs must be associated with the exposure of interest (the relevance assumption). Second, IVs must not be associated with any confounder in the exposure–outcome relationship (the exchangeability assumption). Third, IVs must not affect the outcome except through the exposure (the exclusion restriction assumption). There are several methods to estimate the causal effect using MR, the simplest being the Wald method, whereby a ratio is taken between the variant–outcome association and the variant–risk factor association. A natural extension of this approach, known as the inverse-variance weighted (IVW) method, combines multiple IVs, applied in this report[37]. The causal effect of exposure $X$ on the outcome $Y$, using $k$ genetic variants, is given by

$$\hat{\alpha} = \frac{\sum_k \beta_k^X \beta_k^Y (\sigma_k^Y)^{-2}}{\sum_k (\beta_k^X)^2 (\sigma_k^Y)^{-2}}$$

with the corresponding variance $\mathrm{Var}(\hat{\alpha}) = \frac{1}{\sum_k (\beta_k^X)^2 (\sigma_k^Y)^{-2}}$, where $\beta_k^X$ and $\beta_k^Y$ represent the estimated effects of genetic variant $k$ on $X$ and $Y$, respectively, and $\sigma_k^Y$ represents the standard error of $\beta_k^Y$. Here, we extended the framework of MR to situations where the exposure and the outcome are measured in different individuals. More specifically, our exposure was a trait of an index individual, and the outcome was the same (or another) trait in its partner.

### Phenotype selection and processing

We used an agnostic, phenome-wide approach for selecting phenotypes. Specifically, we first selected phenotypes which were analysed by the Neale group and which had men, women and joint summary statistics available (http://www.nealelab.is/uk-biobank/). This list was intersected with our internal database (application number 16389), leaving 1,278 phenotypes available for analysis. Phenotypes were processed in the filtered QC dataset ($n = 337{,}138$) according to a slightly modified version of the PHEnome Scan ANalysis Tool (PHESANT) pipeline to accommodate the phenotypes that we had available in our database[38]. Continuous variables were transformed to a normal distribution using a rank-preserving inverse normal quantile transformation, while ordinal and binary traits were recategorized according to PHESANT documentation (for example, categories with fewer than ten participants were removed). We then filtered these phenotypes as follows. First, to focus on traits with some indication of assortment, we computed the raw phenotypic correlation ($r_{x_i x_p}$, where $x$ refers to trait $X$, i the index person and p is its partner) among couples and removed phenotypes with a Pearson correlation of <0.1. To ensure that inverse normal quantile transformation was not significantly impacting the correlations of each trait, we also calculated the correlation between partners for each trait using the nonparametric Spearman correlation and found consistent estimates (Supplementary Fig. 2). Second, we removed phenotypes that had fewer than five valid IVs for MR. IVs were defined based on an association $P < 5 \times 10^{-8}$ in the joint Neale summary statistics, after pruning for independence (based on a clumping procedure performed in PLINK[39] with the options

–clump-kb 10000 and –clump-r2 0.001 using the 1000 Genomes European samples as a reference). Third, using the sex-specific summary statistics, the IV heterogeneity between sexes was calculated. IVs that showed (Bonferroni-corrected) significant evidence of heterogeneity between sexes were excluded ($P < 0.05/$[number of IVs]). After this procedure, phenotypes were again filtered to those with at least five valid IVs remaining. Fourth, dietary phenotypes were removed due to high correlation among these phenotypes (due to the shared household), insufficient power, problems with reverse causation and difficult interpretation[40]. Finally, we manually removed several duplicated and redundant phenotypes. Specifically, (1) left-side body traits (highly correlated with right side) were removed; (2) we retained only one of the duplicated phenotypes for BMI and weight (retaining UKBB data fields 21001 and 21002, respectively); and (3) all 'qualifications' data were removed (corresponding to UKBB field 6,138) due to the availability of finer-scale correlated variables, such as 'age completed full-time education' (data field 845). This ad hoc procedure was meant to capture only major redundancies. After this process, 118 phenotypes remained for analysis (Supplementary Fig. 3).

### Estimation of single-trait causal effects in couples

To investigate the causal effect of a trait in one individual on the same trait of their partner, we performed couple-specific MR analyses. Specifically, the trait in the index case was used as the exposure and the same trait in the partner was used as the outcome trait. The effect of genetic variants on the exposure was obtained from the Neale summary statistics using the full UKBB sample. Instruments for each trait were selected as described above: that is, being both genome-wide significant ($P < 5 \times 10^{-8}$) and pruned for independence. Next, we estimated the effects of single-nucleotide polymorphisms (SNPs) on the outcomes of interest by testing the association between each genetic instrument measured in the index individual with the phenotype measured in the partner using the UKBB partner dataset described above. In other words, for each phenotype, the corresponding genetic data for the IVs were obtained from the index case, while the phenotypes (dependent variable) were taken from the corresponding partner. All SNP trait estimates were estimated in men and women separately (that is, using the sex-specific Neale summary statistics or two separate models in the couple data), adjusting for age and the first 40 genetic principal components of both the index and partner. To mimic the Neale models, we performed linear regression of SNP effects on phenotypes, regardless of data type (including binary). Continuous phenotypes were scaled to have mean of zero and s.d. of one before regression, while ordinal and binary phenotypes were left as processed by PHESANT.

To estimate the causal effect of a trait from an index case to a partner ($\alpha_{x_i \rightarrow x_p}$), we combined the effects of genetic instruments on the exposure (from Neale) with effects on the outcomes (measured among couples) in an MR framework using the IVW method (Fig. 2a)[37]. To estimate the causal effects in both sexes combined, SNP effects were first meta-analysed across sexes using fixed effects models before performing MR (rather than meta-analysing the MR estimates directly) to minimize weak instrument bias[41]. Effects of the genetic estimates on both the exposure and outcome were first standardized (such that the squared effect size represents the explained variance) to allow for seamless comparison across traits and to the raw phenotype correlation. Significance was determined by adjusting for the number of effective tests based on the correlation matrix of phenotypes tested[42], resulting in 66 independent tests. The significance threshold was adapted accordingly as $P < 0.05/66$.

After estimating single-trait causal effects in couples, we used a two-tailed $Z$ test to identify traits with a significant difference between the MR estimate and the phenotypic correlation in couples. For each trait with discrepant estimates, we tested the causal effect of each of the remaining 117 phenotypes in our pipeline ($Y_1, \ldots, Y_{117}$) on the focal trait of interest ($X$) using MR ($\alpha_{y \rightarrow x}$). These same-person MR estimates

were calculated using meta-analysed sex-specific Neale estimates for both the SNP exposure and SNP outcome effects using the IVW method. Before performing each same-person MR, genetic variants were first filtered for evidence of reverse causality at a threshold of $P < 0.001$ (Steiger filter)[43], whereby SNPs were removed if the standardized SNP effect on the outcome was stronger than the effect on the exposure based on a one-tailed $t$ test at a significance level of $P < 0.001$. SNP effects were standardized before calculating MR effects.

We then explored those potential confounders, $Y_1, Y_2, \ldots, Y_{117}$, with a significant impact on $X$ ($P < 0.05/66$). As the confounding impact of each $Y_k$ involves a within-couple effect ($\alpha_{y_i \to y_p}$), as illustrated in Fig. 2b, we further filtered the remaining $Y_k$ traits to those with a significant within-couple MR effect ($P < 0.05/$[number of remaining $Y_k$]). The couple correlation induced by confounder $Y$ can be expressed as $\alpha^2_{y \to x} \times \alpha_{y_i \to y_p}$. We restricted confounders to only those that had a correlation with $X$ less than 0.8 to avoid using meaninglessly similar traits to $X$. Since potential confounders may be correlated, one could use MVMR to assess the joint contribution of these confounders on the couple correlation for trait $X$. This, however, led to numerical instability, and we decided to rather prune confounders ($r^2 < 0.1$, prioritizing for larger $\hat{\alpha}_{y \to x}\hat{\alpha}_{y_i \to y_p}$ values) and obtain $\hat{\alpha}_{y \to x}$ estimated from univariable MR, where IVs were pruned for independence ($r^2 < 0.001$). Finally, we plugged in the obtained causal effect estimates for the $m$ remaining confounder traits ($Y_1, Y_2, \ldots, Y_m$) on $X$, ($\hat{\alpha}_{Y_1 \to X}, \hat{\alpha}_{Y_2 \to X}, \ldots, \hat{\alpha}_{Y_m \to X}$), into the estimator of the correlation induced by these confounders to get the estimate of total confounding $C = \sum_{j=1}^{m} \hat{\alpha}_{(Y_j)_i \to (Y_j)_p}(\hat{\alpha}_{Y_j \to X})^2$. The variance of such a sum was estimated as the sum of the variances of the individual terms (since they are uncorrelated). The variance of $(\hat{\alpha}_{Y_j \to X})^2$ was approximated by assuming $\hat{\alpha}_{Y_j \to X}$ following a normal distribution and used the general formula of $\mathrm{Var}(X^2) = 4\mu^2\sigma^2 + 2\sigma^4$ for $X \approx N(\mu, \sigma^2)$. The variance of the product of $\hat{\alpha}_{(Y_j)_i \to (Y_j)_p}$ and $(\hat{\alpha}_{Y_j \to X})^2$ was estimated based on the formula for the variance of the product of independent random variables: $\mathrm{Var}(XY) = \mathrm{Var}(X)\mathrm{Var}(Y) + \mathrm{Var}(X)E^2(Y) + \mathrm{Var}(Y)E^2(X)$.

## The role of confounders on trait correlation in couples

We sought to explore the impact of potential confounders on mate choice by calculating the trait correlations between partners that are due to confounding. We considered the impact of the following confounders ($Y$) on the partner correlations of the remaining 117 traits selected by our pipeline: average household income, age completed full-time education, sports club or gym user, current tobacco smoking, overall health rating, and north and east birth place coordinates (UKBB data fields 738; 845; 6160; 1239; 2178; 129; and 130, respectively). The choice of these traits was somewhat ad hoc, mostly driven by previous evidence for driving trait similarities and themselves showing strong couple correlation in the UKBB. Using the single-trait causal effects in couples and the same-person MR estimates, correlation due to founding was calculated for each pair ($Y, X$) as $C = \alpha^2_{y \to x}\alpha_{y_i \to y_p}$ (Fig. 2b). These confounding estimates were finally contrasted to the couple correlation values to explore the extent that each $Y$ may confound couple correlations by examining the ratio between the two estimates (that is, $\frac{C}{\mathrm{cor}(X_i, X_p)}$). Birthplace coordinates (east–west and north–south) were considered together and their invoked trait correlations were summed up, as they are orthogonal by definition.

## Trait convergence over time

Trait similarity in couples can be driven by both mate choice and/or trait convergence over time spent together. To tease out the contribution of these different sources, we explored whether the cross-partner causal effects change as a function of the length of the relationship and age. The length of relationship was proxied by the minimum value of the 'length of time at current address' (data field 699) for the two partners. To estimate the effect of age, we took the median age of couples. For each of the two derived variables, we split the couples into five roughly

equal-sized bins (using the 'smart_cut' function from the *cutr* R package). We first estimated the phenotypic correlation of each trait within couples of each bin. Next, for each single-trait MR described above, analyses were run in the full sample as well as in the different bins. Of the significant results identified in the sex-combined analysis above, we tested to see if there was any significant difference in MR estimates among the bins. Binned MR estimates were computed using the SNP outcome effect estimated in each bin separately, and the SNP outcome effects used the same SNP exposure effects from Neale. Analyses were run in each sex separately and combined (meta-analysed at the SNP level). As above, SNP effects were standardized before calculating MR estimates. To assess for the presence of a trend across bins, we tested the significance of the slope of a linear model of bin-specific correlations and bin-specific MR estimates, inversely weighted by the SE, versus the bin centre (that is, the median age or time spent together for the given bin). Multiple testing was, as described above, adapted based on the effective number of tests but restricted to traits which showed significant causal effects in the joint (both sexes combined), nonbinned MR (resulting in a threshold of $P < 0.05/29$).

## Estimation of cross-trait causal effects in couples

Using the same process as in the AM analysis involving a single trait, we also sought to investigate causal effects within couples involving two traits ($\alpha_{x_i \to y_p}$). In other words, two different traits were used as exposure and outcome to determine the causal effects of trait $X$ (in the index individual) on trait $Y$ (in the partner). Here, we only considered trait combinations with phenotypic correlation of <0.8 (estimated in the entire UKBB, $n = 337{,}138$) to avoid too closely related traits. The same set of SNPs was used as in the same-person MR (that is, first filtered for the presence of reverse causality). As in the single-trait MR, SNP exposure effects were obtained from the Neale summary statistics and SNP outcome effects were estimated in the couple-derived dataset. MR models were run in both sexes separately and jointly (meta-analysing the SNP effects before performing MR analyses). Significance was determined based on the squared effective number of tests ($P < 0.05/[66^2]$).

## Comparison of paths from index to partner

There are several independent paths through which a trait in an index case could exert a causal effect on another trait in the partner. We wanted to explore if one path was more dominant, in general, and if there was evidence for the presence of direct effects (or indirect effects with additional traits involved). Restricting to Bonferroni-significant trait pairs (with phenotypic correlation of <0.8) from the couple MR, we sought to explore the various paths through which a phenotype $X$ in an index case ($X_i$) could causally impact a phenotype $Y$ in the partner ($Y_p$) as illustrated in Fig. 2c. With the exception of exposure traits that directly alter the environment of their partner, such as smoking creating the presence of secondhand smoke, $X_i$ is unlikely to have a direct effect on another $Y_p$. Alternatively, $X_i$ might indirectly impact $Y_p$ by inducing changes in $X_p$, which in turn, impacts $Y_p$. For instance, increased BMI in an index case is not expected to directly increase cardiovascular disease risk in their partner but rather, to modify the partner's risk through first increasing their BMI. To explore this intuition, we dissected the causal effect from $X_i$ to $Y_p$ ($\omega$) into three possible (nonindependent) mechanisms. First, $X_i$ could exert a causal effect on $X_p$, followed by $X_p$ having a causal effect on $Y_p$ in the partner alone ($\gamma$). Second, the reverse could occur, whereby $X_i$ has a causal effect on $Y_i$ in the index alone, followed by a causal effect of $Y_i$ case on $Y_p$ ($\rho$). Third, there could be other mechanisms, either acting directly or through other unmeasured/considered variables. These three scenarios could also act in some combination. In this way, the $\omega$ estimate would capture the paths of $\gamma$, $\rho$ and other mechanisms combined.

Using the same-person MR estimates that were calculated as described above, we estimated $\gamma$ and $\rho$ representing the various paths

from $X_i$ to $Y_p$ and contrasted them to the total cross-trait cross-partner effect $\omega$. To quantify $\gamma$, the single-trait couple causal effect estimate (that is, from the regression $X_p \sim X_i$) was multiplied by the same-individual causal estimate (that is, $\alpha_{x \to y}$ from $Y \sim X$). To quantify $\rho$, we first estimated the causal effect of $Y_i$ on $Y_p$ in MVMR to exclude any residual effect of $X$ on phenotype $Y$ from index to partner. Specifically, $Y_p$ was used as the independent variable with both $Y_i$ and $X_i$ as independent variables (that is, the MVMR was $Y_p \sim Y_i + X_i$). We included both IVs from $X$ and $Y$, pruned for independence (performed in PLINK with the options –clump-kb 10000 and –clump-r2 0.001 using the 1000 Genomes European samples as a reference). We took the coefficient of $Y_i$ as the direct causal effect from $Y_i$ to $Y_p$ ($\alpha_{y_i \to y_p}$) and multiplied this by the same-individual causal estimate ($\alpha_{x \to y}$). Finally, we estimated $\omega$ directly from our cross-trait couple MR framework ($\alpha_{X_i \to y_p}$). We compared the estimates of $\gamma$, $\rho$ and $\omega$ using a $Z$ test to assess their difference and using linear regression with the intercept forced through the origin to determine their relationship. Finally, we quantified the proportion of $\omega$ that could not be explained by the paths quantified by $\gamma$ and $\rho$. As $\gamma$ and $\rho$ are not perfectly independent, potentially due to correlation between $X$ and $Y$ or pleiotropic limitations of MR, we estimated the extent of dependence via the correlation between $\rho$ and $\gamma$ across the different trait pairs. To account for the duplicate signals due to this correlation, we removed the effects of $\gamma$ from $\rho$ by keeping the residuals from the linear regression $\rho \sim \gamma$. We then estimated the proportion of variance explained ($R^2$) of $\omega$ jointly by $\gamma$ and the residualised $\rho$.

### Biases in the causal effect estimation

Violations of the MR assumptions are different in applications like ours, where the exposure and outcome are the same trait but measured in different individuals. Hartwig et al.[44] explored the impact of AM on MR but for the classical setting of testing different traits in the same individual. Therefore, we set up a model (Fig. 7) specific to causal effect estimations where exposures and outcomes are the same traits but in different individuals (couples) to examine the potential issues. We have allowed for a more complex set of MR violations, which we describe below.

The model focuses on trait $X$ in the offspring (O) ($X_O^i$) and that in the partner ($X_O^p$). Analogously, we denoted the same trait in the offspring's mother and father with subscripts M and F. These traits are influenced by direct genetic effects ($G_O^i$) and direct (non-genetic) environmental effects ($E_O^i$), with effect sizes $g$ and $e$, respectively. Everything derived below works the same way even if $G_O^i$ represents a single SNP (and hence, $E_O^i$ is heritable). The offspring environment is influenced by the parental environment ($E_P^i$), the parental genes ($G_P^i$) and the parental trait ($X_P^i$). Parental characteristics ($G, E, X$) are simply defined as a rescaled average of the maternal and paternal traits: that is, $X_P^i = (X_M^i + X_F^i)/\sqrt{2(1 + \mathrm{cor}(X_M^i, X_F^i))}$ to ensure that the trait variance is equal to 1, i.e. $\mathrm{Var}(X_P^i) = 1$. Note that this simplification assumes that paternal and maternal effects are identical, which holds in general[32]. The direct causal effects of $G_P^i$, $E_P^i$ and $X_P^i$ on $E_O^i$ are denoted by $s_G$, $s_E$ and $s_X$, respectively. By definition, the correlation between mean parental genotype and offspring genotype is $1/\sqrt{2}$. Parental traits ($E_P^i, X_P^i$) cannot modify offspring genotype; thus, they can influence $X_O^i$ only via its environmental component ($E_O^i$). The same description holds for all corresponding variables of the partner. Finally, $X_O^i$ and $X_O^p$ are paired such that predefined correlations between $G, E$ and $X$ are satisfied. For this, direct assortment coefficients $r_G, r_E$ and $r_X$ are defined between $G, E$ and $X$, respectively.

As can be seen from Fig. 7, parental effects induce a correlation of $\left(\frac{1}{\sqrt{2}}\right)(s_G + s_X \times g)$ between the offspring genotype ($G_O^i$) and the offspring environment ($E_O^i$). Also, one can note that the total couple correlation for $X$ can arise from three independent sources: direct $X$ assortment ($r_X$), through direct assortment for $E$ ($r_E \times e^2$) and through direct assortment through $G$ ($r_G \times g^2$). Thus, the expected correlation between $X_O^i$

and $X_O^p$ is $r_X + r_E \times e^2 + r_G \times g^2$. Similarly, the expected correlation between $E_O^i$ and $E_O^p$ is $r_E + r_X \times e^2 + r_G \left(\left(\frac{1}{\sqrt{2}}\right)(s_G + g \times s_X)\right)^2$ and between $G_O^i$ and $G_O^p$ is $r_G + r_X \times g^2 + r_E \left(\left(\frac{1}{\sqrt{2}}\right)(s_G + g \times s_X)\right)^2$.

Under this model, the expected effect of the index genotype on the index trait is

$$E\left[\hat{\beta}_{G^i \to X^i}\right] = g + \left(\frac{1}{\sqrt{2}}\right)(s_G + g \times s_X)\,e,$$

while the expected effect of the index genotype on its partner's trait is

$$E\left[\hat{\beta}_{G^i \to X^p}\right]$$
$$= \left(g + \left(\frac{1}{\sqrt{2}}\right) \cdot (s_G + g \times s_X)\,e\right)r_X + r_G \times g + \left(\frac{1}{\sqrt{2}}\right)(s_G + g \times s_X)(r_E \times e).$$

Therefore, the expectation of the estimated causal effect $X_O^i \to X_O^p$ can be written as

$$E\left[\alpha_{X_O^i \to X_O^p}\right] \approx \frac{E[\hat{\beta}_{G^i \to X^p}]}{E[\hat{\beta}_{G^i \to X^i}]} = r_X + \frac{r_G \times g + \left(\frac{1}{\sqrt{2}}\right)(s_G + g \times s_X)(r_E \times e)}{g + \left(\frac{1}{\sqrt{2}}\right)(s_G + g \times s_X)\,e}.$$

### Several parameters in this model can lead to the violation of MR assumptions

- Violation 1 ($s_G \neq 0$): this violation implies that parental genetics directly impact offspring environment (not through $E$ or $X$). Such violation would most likely happen if $G_P^i$ impacts another parental phenotype $Y_P^i$ which has an impact on $E_O^i$: that is, $s_G = g \cdot r_G(X, Y)\,s_Y$. Here, $r_G(X, Y)$ refers to the genetic correlation between traits $X$ and $Y$. The formula assumes that the effect of $G_P^i$ to a secondary trait ($Y$) is expected to be its genetic effect on the primary trait ($X$) multiplied by their genetic correlation.

- Violation 2 ($s_X \neq 0$): this violation is the classical parental rearing (or in other terms, dynasty) effect.

- Violation 3 ($s_E \neq 0$): this violation does not lead to MR bias since there is no path from the offspring genotype to the offspring environment through the parental environment.

- Violation 4 ($r_G \neq 0$): this violation allows direct assortment based on index and partner genetics, which most likely happens due to assortment for another trait ($Y$). Hence, its typical size is $r_G = (g \times r_G(X, Y))^2 r_Y$. Here, we made a similar assumption as for Violation 1. One (toy) example for such scenario could be educational attainment ($X$), whose genetic component is strongly associated with intelligence, for which additional assortment occurs.

- Violation 5 ($r_E \neq 0$): this violation allows direct assortment based on the environment of the index and partner. A simplified example for this scenario is trait $X$ being BMI and $E$ representing 'going to the gym'. In such case, beyond AM for BMI, people are more likely to choose their partners from the gym (selecting for fitter individuals); hence; the environment has an additional direct effect of the partner's BMI.

To gauge the extent of the MR bias in realistic parameter settings, we have visualized the bias for a wide range of parameters. We set $r_X = 0.2$, the heritability to 20% (hence, $g = \sqrt{0.2}$ and $e = \sqrt{1 - 0.2}$). Then, two of the remaining four crucial parameters ($s_G, s_X; r_E, r_G$) were fixed and the other two were varied. For example, the chosen range for $r_G$ assumed a value of $r_G(X, Y) = 0.3$ and $r_Y$ range of $[-0.3, 0.3]$ using the abovementioned formula of $r_G = (g \times r_G(X, Y))^2 r_Y$. Similarly, we used the formula $s_G = g \times r_G(X, Y)\,s_Y$ to justify the explored range for $s_G$, with $r_G(X, Y) = 0.3$ and $s_Y$ between −0.3 and 0.3.

## Reporting summary

Further information on research design is available in the Nature Portfolio Reporting Summary linked to this article.

## Data availability

The UK Biobank data are available through the standard UK Biobank application procedure (https://www.ukbiobank.ac.uk/enable-your-research/apply-for-access). The household information can be separately requested from the UK Biobank access team, as it is not part of the variables listed on the showcase (https://biobank.ndph.ox.ac.uk/showcase/index.cgi). The 1000 Genomes European genetic data can be downloaded from http://ftp.1000genomes.ebi.ac.uk/vol1/ftp/.

## Code availability

Custom code is available at https://github.com/jennysjaarda/proxyMR.

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

## Acknowledgements

For computations, we used the CHUV HPC cluster. We thank C. Auwerx, M. Sadler and A. van der Graaf for their valuable feedback on the manuscript. This research has been conducted using the UK Biobank Resource under Application Number 16389. Z.K. was funded by the Swiss National Science Foundation (310030-189147). The funders had no role in study design, data collection and analysis, decision to publish or preparation of the manuscript.

## Author contributions

Z.K. devised and directed the project. J.S. and Z.K. contributed to the mathematical derivations, design and implementation of the research; to the analysis of the results; and to the writing of the manuscript.

## Funding

## Competing interests

The authors declare no competing interests.

## Additional information

**Correspondence and requests for materials** should be addressed to Zoltán Kutalik.

# Reporting Summary

## Statistics

For all statistical analyses, confirm that the following items are present in the figure legend, table legend, main text, or Methods section.

| n/a | Confirmed | |
|---|---|---|
| ☐ | ☒ | The exact sample size (*n*) for each experimental group/condition, given as a discrete number and unit of measurement |
| ☐ | ☒ | A statement on whether measurements were taken from distinct samples or whether the same sample was measured repeatedly |
| ☐ | ☒ | The statistical test(s) used AND whether they are one- or two-sided *Only common tests should be described solely by name; describe more complex techniques in the Methods section.* |
| ☐ | ☒ | A description of all covariates tested |
| ☐ | ☒ | A description of any assumptions or corrections, such as tests of normality and adjustment for multiple comparisons |
| ☐ | ☒ | A full description of the statistical parameters including central tendency (e.g. means) or other basic estimates (e.g. regression coefficient) AND variation (e.g. standard deviation) or associated estimates of uncertainty (e.g. confidence intervals) |
| ☐ | ☒ | For null hypothesis testing, the test statistic (e.g. *F*, *t*, *r*) with confidence intervals, effect sizes, degrees of freedom and *P* value noted *Give P values as exact values whenever suitable.* |
| ☒ | ☐ | For Bayesian analysis, information on the choice of priors and Markov chain Monte Carlo settings |
| ☒ | ☐ | For hierarchical and complex designs, identification of the appropriate level for tests and full reporting of outcomes |
| ☐ | ☒ | Estimates of effect sizes (e.g. Cohen's *d*, Pearson's *r*), indicating how they were calculated |

*Our web collection on statistics for biologists contains articles on many of the points above.*

## Software and code

Policy information about availability of computer code

| Data collection | No software was used for data collection. |
|---|---|
| Data analysis | Data analysis was performed using code written in the freely available R programming language (R version 3.6.3 (2020-02-29)) and is available at https://github.com/jennysjaarda/proxyMR. The cutr package was downloaded from https://github.com/moodymudskipper/cutr. |

For manuscripts utilizing custom algorithms or software that are central to the research but not yet described in published literature, software must be made available to editors and reviewers. We strongly encourage code deposition in a community repository (e.g. GitHub). See the Nature Portfolio guidelines for submitting code & software for further information.

## Data

Policy information about availability of data

All manuscripts must include a data availability statement. This statement should provide the following information, where applicable:

- Accession codes, unique identifiers, or web links for publicly available datasets
- A description of any restrictions on data availability
- For clinical datasets or third party data, please ensure that the statement adheres to our policy

The UK Biobank data is available through the standard UK Biobank application procedure (https://www.ukbiobank.ac.uk/enable-your-research/apply-for-access). The household information can be separately requested from the UK Biobank access team, as it is not part of the variables listed on the showcase (https://biobank.ndph.ox.ac.uk/showcase/index.cgi). The 1000 Genomes European genetic data can be downloaded from http://ftp.1000genomes.ebi.ac.uk/vol1/ftp/. Simulated data can be produced by executing the provided code.

# Field-specific reporting

Please select the one below that is the best fit for your research. If you are not sure, read the appropriate sections before making your selection.

☒ Life sciences          ☐ Behavioural & social sciences          ☐ Ecological, evolutionary & environmental sciences

For a reference copy of the document with all sections, see nature.com/documents/nr-reporting-summary-flat.pdf

# Life sciences study design

All studies must disclose on these points even when the disclosure is negative.

| | |
|---|---|
| Sample size | All available (N=51,664) couples in the UK Biobank after exclusion descirbed in Supplementary Figure 1. |
| Data exclusions | Non-British individuals, or those with missing data were excluded from the analysis. Further exclusion criteria are fully described in Supplementary Figure 1. |
| Replication | No sufficiently large other data set was available for replication. All analysis results can be reproduced based on the provided code. |
| Randomization | We had no experimental groups. |
| Blinding | There was no group allocation. |

# Reporting for specific materials, systems and methods

We require information from authors about some types of materials, experimental systems and methods used in many studies. Here, indicate whether each material, system or method listed is relevant to your study. If you are not sure if a list item applies to your research, read the appropriate section before selecting a response.

### Materials & experimental systems

| n/a | Involved in the study |
|---|---|
| ☒ | ☐ Antibodies |
| ☒ | ☐ Eukaryotic cell lines |
| ☒ | ☐ Palaeontology and archaeology |
| ☒ | ☐ Animals and other organisms |
| ☐ | ☒ Human research participants |
| ☒ | ☐ Clinical data |
| ☒ | ☐ Dual use research of concern |

### Methods

| n/a | Involved in the study |
|---|---|
| ☒ | ☐ ChIP-seq |
| ☒ | ☐ Flow cytometry |
| ☒ | ☐ MRI-based neuroimaging |

## Human research participants

Policy information about studies involving human research participants

| | |
|---|---|
| Population characteristics | Described in the UK Biobank protocol: https://www.ukbiobank.ac.uk/wp-content/uploads/2011/11/UK-Biobank-Protocol.pdf |
| Recruitment | See above. |
| Ethics oversight | This study used the UK Biobank (UKBB) cohort, a prospective population-based study with over 500,000 adult participants. UK Biobank has approval from the North West Multi-centre Research Ethics Committee (MREC) as a Research Tissue Bank (RTB) approval. This approval means that researchers do not require separate ethical clearance and can operate under the RTB approval. UK Biobank also possesses a Human Tissue Authority (HTA) licence, so a separate HTA licence is not required by researchers who receive samples from the resource. |

Note that full information on the approval of the study protocol must also be provided in the manuscript.

