## [Peer Review File · Nature Human Behaviour]

Peer Review Information

Journal: Nature Human Behaviour

Manuscript Title: Partner choice, confounding and trait convergence all contribute to phenotypic partner similarity

Corresponding author name(s): Zoltán Kutalik

Reviewer Comments & Decisions:

Decision Letter, initial version:

1st June 2022

Dear Professor Kutalik,

Thank you once again for your manuscript, entitled "The contribution of mate-choice, couple convergence and confounding to assortative mating", and for your patience during the peer review process.

Your Article has now been evaluated by 3 referees. You will see from their comments copied below that, although they find your work of [considerable] potential interest, they have raised quite substantial concerns. In light of these comments, we cannot accept the manuscript for publication, but would be interested in considering a revised version if you are willing and able to fully address reviewer and editorial concerns.

We hope you will find the referees' comments useful as you decide how to proceed. If you wish to submit a substantially revised manuscript, please bear in mind that we will be reluctant to approach the referees again in the absence of major revisions. We are committed to providing a fair and constructive peer-review process. Do not hesitate to contact us if there are specific requests from the reviewers that you believe are technically impossible or unlikely to yield a meaningful outcome.

To guide the scope of the revisions, the editors discuss the referee reports in detail within the team, including with the chief editor, with a view to (1) identifying key priorities that should be addressed in revision and (2) overruling referee requests that are deemed beyond the scope of the current study. We hope that you will find the prioritised set of referee points to be useful when revising your study. Please do not hesitate to get in touch if you would like to discuss these issues further.

In your revision, we ask that you include additional analyses to demonstrate the validity of your MR analyses under AM, as suggested by Reviewer #1. Please also consider potential measurement errors

and their impact, and provide evidence that your proxies for the relationship length are valid. Crucially, given the comments of Reviewer 1, please also only interpret your results that are statistically significant after correction, and remove the discussion of gender roles as choosers/courters if this is not supported by your analyses.

If you wish to submit a suitably revised manuscript we would hope to receive it within 4 months. I would be grateful if you could contact us as soon as possible if you foresee difficulties with meeting this target resubmission date.

- Include a "Response to the editors and reviewers" document detailing, point-by-point, how you addressed each editor and referee comment. If no action was taken to address a point, you must provide a compelling argument. When formatting this document, please respond to each reviewer comment individually, including the full text of the reviewer comment verbatim followed by your response to the individual point. This response will be used by the editors to evaluate your revision and sent back to the reviewers along with the revised manuscript.
- Highlight all changes made to your manuscript or provide us with a version that tracks changes.

[REDACTED]

Thank you for the opportunity to review your work. Please do not hesitate to contact me if you have any questions or would like to discuss the required revisions further.

Sincerely,

Arunas Radzvilavicius, PhD
Editor
Nature Human Behaviour

Reviewer expertise:

Reviewer #1: statistical genetics, mendelian randomization

Reviewer #2: statistical genetics, assortative mating, mendelian randomization

Reviewer #3: behavioural genetics

REVIEWER COMMENTS:

Reviewer #1:
Remarks to the Author:

The authors present a novel and interesting analyses of assortative mating using Mendelian randomization (MR) that was enjoyable to read. The paper has the potential to make a valuable contribution to the literature, though the authors must further validate their approach as it is known that their method for examining AM (MR) is itself biased by MR. My concerns are as follows:

* Major concerns

- The authors rely on Mendelian Randomization (MR) to examine causality between individuals, in an approach similar to Howe et al (doi.org/10.1038/s41467-019-12424-x). However, it is known that AM violates assumptions necessary for valid causal inference with MR (e.g., [/doi.org/10.1038/s41467-020-17117-4](https://doi.org/10.1038/s41467-020-17117-4)). This is paradoxically noted by Howe et al, who in their discussion state that their MR-based evidence for AM is important because AM violates MR assumptions. I.e., to the extent their findings are correct, they might be incorrect. In both that manuscript and the current manuscript, this issue is not adequately addressed. Extensive simulations would be necessary to convince me that this is a valid approach, particularly considering the following factors:

- presence or absence of genetic correlation between focal traits
- presence or absence of vertical transmission (i.e. non genetic factors transmitted from child to offspring)

As the entirety of the manuscript hinges on the validity of MR (which is known to break under AM), this is a critical missing piece of analysis.

- The notion that the authors put forth prominently in their discussion, that women are choosers and men are courters is far beyond anything supported by their data. They observe no significant differences in causal effects across sexes after Bonferroni correction for 66 tests (which is liberal considering the total number of tests performed across the entire study). They then redo their hypothesis test choosing a more liberal threshold based (29 tests), and still don't finding anything significant. They select only the 15 nominally significant traits and do a t test on these traits only (which is virtually guaranteed to be significant due to the winner's curse) and get $p = .014$. If you keep altering your statistical procedure until you find something, the statistics lose meaning.

* Additional concerns

- (line 54) The authors here suggest that assortative mating will lead concentration of genetic resources. This assertion is not supported by any the three papers they cite and is precisely the

opposite of what they imply. Specifically, AM increases genetic variance, leading to dispersion, not concentration, of "genetic resources". It does not lead to a bimodal distribution. For every pair of individuals at the extremes of the genetic spectrum mating assortatively, many more will be somewhere in the middle.

- Every step of significance-based thresholds for which traits to consider in subsequent analyses induces winner's curse bias--I'd encourage the authors to analyze all traits. Also, perhaps one reason they didn't find much evidence for suppression effects is that they didn't examine confounding factors for traits without significant direct effects

- I'd encourage the authors to rethink or supplement their use of the phrase "confounding" with alternative language or further discussion. Though it makes perfect sense in the context of causal analysis, it has the unfortunate connotation of implying that an observed association is spurious. This is relevant in light of recent evidence that simultaneous on multiple phenotypes can have more dramatic consequences for genetic architecture / inference. So even if a cross-mate association can be partly accounted for via an indirect path, this doesn't mitigate its impact on quantities of interest (and could even make them worse). I don't think the authors make this mistake, but their readers might.

- The results about spousal convergence are not particularly compelling. First, time at shared address is a weak proxy for time spent together as many couples will have lived together prior to moving to their current residence (unless I'm misunderstanding?). Second, given the lack of age diversity in the UKB, there are few young couples. Third, the phenotypes for which spousal convergence is likely the strongest (income) is measured at the couple level (household), not the individual level.

- the authors identified a larger number of spouses in the UKB than others have previously (e.g., Howe et al, Yengo et al [doi.org/10.1038/s41562-018-0476-3], Border et al [doi.org/10.1101/2022.03.21.485215]). It would be interesting to see which criteria are responsible for these differences.

- (line 234) typo, I believe $\alpha_{z \rightarrow x}$ should be $\alpha_{y \rightarrow x}$, also a Z on line 273 should be a Y, unless I'm confused

- (line 500) typo, "Figure 1C" should be 2C

* Additional questions

- Given that many confounders were identified for some traits, it seems likely that some of these confounders are themselves confounded. In (line 444), the authors note that they cannot simply add together the correlations across confounders, but it isn't clear why these effects cannot be jointly estimated. It would be quite interesting to know if individual factors stand out (e.g., you could imagine SES accounting for a large amount of confounding signal observed for BMI/diet related traits) or if it's more of a wash (e.g., perhaps the confounding signals seen across metabolic phenotypes are largely overlapping).

Reviewer #2:

Remarks to the Author:

Sjaarda and Kutalik present very interesting research using genetic and phenotype data to tease apart why spouses are similar. I think this is an important research question as spouses are correlated for nearly every phenotype and often this does not relate to mate-choice on the index phenotype but rather to assortment on other phenotypes such as educational attainment. For example, spouses are correlated for many biomarkers which as non-visible phenotypes are very unlikely to influence mate choice. The results presented here are very compelling evidence that partner similarities are driven by a smaller set of phenotypes.

I had a few concerns about the language and some of the conclusions detailed below. I am confident that the manuscript will be an important contribution to the field once refined.

1) Semantics of assortative mating.

Title "The contribution of mate-choice, couple convergence and confounding to assortative mating"

Abstract "Increased phenotypic similarity between partners, termed assortative mating (AM), has been observed for many traits."

Introduction "In human populations, phenotypic similarity exists between partners compared to random pairs, a phenomenon known as (positive) assortative mating (AM)."

I personally think that equating 'spousal phenotypic similarities' and 'assortative mating' as synonymous is confusing. To me, assortative mating refers to individuals preferring to select (and perhaps remain in a relationship with) a phenotypically (dis)similar partner (which the authors have referred to as "mate choice"). I think it reads oddly when assortative mating is defined as including post-mating mechanisms such as spousal convergence from partner interaction effects or the shared environment. For example, if a cohabiting spouse-pair are both exposed to asbestos in their house and then develop lung cancer, would this be included under assortative mating?

I much prefer the way the authors have explained the different mechanisms in Figure 1 (below) where assortative mating (defined as mate choice on phenotypes, geography and social factors etc) is distinguished from post-mating mechanisms.

Illustrates a trait (given by the colour blue) which is under assortative mating, either directly (through mate choice) or due to confounding factors such as shared geography, cultural or religious status or socioeconomic measures. Subsequently, this trait may also undergo post-mating convergence which could be due to direct causal influence from one partner on the other (i.e. through imitation or influence) or due to confounding factors such as shared environment.

The authors are not alone in equating assortative mating and phenotypic similarities and so I'm not necessarily requesting that they necessarily change their terminology. I'm curious what the authors think is the most appropriate terminology and whether they think AM and spousal phenotypic similarities are synonymous. I think it would also be good to be consistent with language throughout as Figure 1 seems to define things slightly differently to the introduction and the abstract.

I would personally be inclined to define AM as all mechanisms inducing spousal similarities at pairing, including direct assortment, indirect assortment and assortment on confounders such as geography. Then define the post-mating mechanisms of couple convergence and shared environmental effects separately. Do the authors think this would be appropriate?

2) Direct assortment - Lines 56 to 66.

Direct assortment here would include both mate choice effects and then also partner interaction and shared environmental effects? For example, if BMI does not influence mate choice but couple's BMI converges over time then this would be direct assortment? Perhaps worth expanding on this in the text as I got a bit confused going from the "three underlying phenomena" to direct and indirect assortment.

3) Spouse definitions

In the methods it states that spouses were defined using household data, but it doesn't explain how these households were defined. Please add in extra detail including the data fields used to assign study participants to the same household, e.g. was this based on shared home coordinates? "Within this sample, we retained individuals coming from households with exactly two unrelated, opposite-sex individuals, leaving 108, 898 participants."

Could you please also return the derived spouse data to UKB, I believe this is mandatory so that other researchers can use them. Perhaps add this in data availability?

4) Proxy for relationship length

The authors use length of time at current address as a proxy for relationship length. I'm concerned that this may have limitations as I'm sure many couples do move around. The UKB generation had a relatively easy time on the housing market compared to the generation of today and so many would have bought increasingly larger properties over time.

Could the authors show the distribution of the proxy variable? Does the distribution seem feasible as a proxy for relationship length? The authors do discuss this as a limited proxy.

5) "Our results point to AM being stronger among females compared to males which is consistent with the notion first put forth by Darwin, that females are, in general, choosers, while males are courters". Is an alternative explanation that women have stronger effects on their partner in the relationship rather than a mechanism relating to partner selection? MR of female genotype to male phenotype should pick up the effects of the woman's genotype on their spouse's phenotype. For example, if the woman is lactose intolerant (LCT genotype) and then buys less milk, this would affect the milk consumption of her partner.

6) Measurement error

Phenotypes can be susceptible to measurement error while genotypes (if imputed well) are often measured accurately. How could phenotypic measurement error potentially affect the different comparisons in the manuscript, specifically the comparison between MR estimates and phenotypic correlations?

Simulations would be helpful to understand if measurement error could have affected any of the conclusions. At the very least, I think measurement error should be acknowledged as a limitation.

Minor comments

1) Abstract line 15 - unclear to me what "causal relationship between partners" means, could you redraft using the assortment terminology?

2) Introduction line 47 - would we use an individual's genome to predict their partner's phenotypes? Maybe rephrase to "is correlated with"?

3) Intro line 54 - "ultimately resulting in a concentration of (genetic) resources" isn't very clear what this means, please elaborate.

4) Line 130 - "The random distribution of genetic variants at birth reduces the possibility of

confounding or reverse causation” Germline genotype being fixed from birth relates to reverse-causation being unlikely, suggest rephrasing.

5) Is it worth also including spousal genetic correlations for comparison? Genetic correlations cannot be impacted by post-mating mechanisms so can only reflect mate choice.

Reviewer #3:

Remarks to the Author:

In this paper, the authors use mendelian randomization to test for causality in assortative mating for 118 traits in a UKB sample of ~52k couples. They find that about half of these traits produced evidence consistent with a causal relationship between partners, and that income represented the largest confounder in cross-partner correlations. They additionally find little evidence for couples growing more similar over time.

This is a nice paper, with a comprehensive set of phenotypes and a big UKB sample. The authors find some interesting effects that align well with my own understanding of existing literature in this area. In sum, this research fills an important gap in our understanding of the context and underlying mechanisms of assortative mating, and I recommend it for publication in NHB. Relatively minor editorial suggestions for the authors follow.

Introduction

In your introductory coverage of various phenotypes that we know to represent AM, I would recommend including religious and political attitudes/values—these in fact represent some of the very strongest AM traits that we know of. See for example a recent adoption study by Willoughby, Giannelis, Ludeke, Klemmensen, Norgaard, Iacono, Lee and McGue (2021) which found enormous spousal correlations for political orientation and related phenotypes as well as religiousness. Earlier studies looking at AM for social attitudes include Kandler, Bleidorn & Riemann (2012) and Eaves et al. (1999).

Apropos of the above studies, it might also be worth briefly discussing an implication of strong assortative mating for some phenotypes: that AM directly influences ACE estimates, which may lead to (e.g.) increasing or decreasing religious and political polarization over time.

Laurence Howe and colleagues also had a paper recently (2021) looking at genetic effects of AM in the UKB, perhaps worth citing in the context of the introduction and subsequent discussion of AM at the genomic level.

Lines 82-83: “(whereas classical MR designs involve a single individual, e.g. BMI risk on CAD).” The acronym “CAD” is not explained in the text and might not be known to readers unfamiliar with medical phenotypes.

Overall I think there are too many parentheticals in the introduction—its last sentence for example (lines 91-93) has two that are nested.

Methods

Some of the coding-related details might be better suited to the supplement. Lines 116-118, for example, would read much better if translated to plain English (e.g., "individuals exhibiting aneuploidy were not included in the final sample"). It's also unclear what "excess relatives" means in that variable name.

Lines 267-268: How/why were these traits chosen for potential cofound variables? I assume from supp table 1 that it's because these traits are all in the top 10 for AM, but then why not religious group? It would be good to see in the main text if there's some other a priori reason for these choices.

Lines 283-284: Using "length of time at current address" as a direct proxy for length of relationship seems potentially problematic: This would not account for those couples wherein one partner moves in with another at their existing address. What was done in cases where the "length of time at current address" differed between two partners in a couple?

Results

It would be interesting to see the results broken down by trait category. Most of the traits studied fall into one of at least three clearly-defined categories, anthropometric traits (BMI, metabolism, appendage fat mass), behavioral traits (education, income, video games, etc), and lifestyle traits (smoking, alcohol, and so on). This intuitive grouping is also referenced in the discussion, with a claim that these groups differed by statistical power (lines 627-630), but readers can't easily confirm that this is true without any actual grouping variable to reference.

Supplement

My only major comment for the supplement is to recommend including a table that includes all 118 phenotypes and their IVW statistics, rather than just the causal-significant ones. This is something readers will definitely be interested in and it would be especially useful for researchers to have a singular reference for all 118 traits.

It would also be good for replication and etc purposes to see the sample size for each trait in supp table 1.

Minor picky comments:

It's really interesting to go through the IVW values for all of the sig traits (supp table 1). But I do have a few questions about some of the trait filtering decisions. In the methods it's stated that redundant phenotypes were removed. It's not clear to me how "(home and birth) location (north coordinate) and "location (east coordinate)" are not redundant—would it not make sense to use one measure of "home location" and one for "birth location"? How should the reader interpret the fact that AM for "north coordinate" is higher (eyeballing it, probably significantly higher) than for "east coordinate"? Does this correspond to the fact that the UK is longer than it is wide, so more variability north-to-south?

Is there any indicator for handedness in the UKB? Reducing the phenotypes to right-body traits makes sense, but I do wonder if handedness would make a difference. AM for arm fat mass (e.g.) might look different if traits were collapsed to "arm fat mass (side corresponding to handedness)".

Author Rebuttal to Initial comments

We would like to thank all three reviewers for their insightful and very valuable comments to improve the manuscript. Below we address each comment point-by-point. Our answers are indicated in **bold blue font**. Quoted texts from the manuscript are “*between quotes and in bold+italic font*”. Please note that we were forced to cut to abstract to 150 words due to journal requirement.

Reviewer #1:

Remarks to the Author:

The authors present a novel and interesting analyses of assortative mating using Mendelian randomization (MR) that was enjoyable to read. The paper has the potential to make a valuable contribution to the literature, though the authors must further validate their approach as it is known that their method for examining AM (MR) is itself biased by MR. My concerns are as follows:

* Major concerns

- The authors rely on Mendelian Randomization (MR) to examine causality between individuals, in an approach similar to Howe et al (doi.org/10.1038/s41467-019-12424-x). However, it is known that AM violates assumptions necessary for valid causal inference with MR (e.g., doi.org/10.1038/s41467-020-17117-4). This is paradoxically noted by Howe et al, who in their discussion state that their MR-based evidence for AM is important because AM violates MR assumptions. I.e., to the extent their findings are correct, they might be incorrect. In both that manuscript and the current manuscript, this issue is not adequately addressed. Extensive simulations would be necessary to convince me that this is a valid approach, particularly considering the following factors:

- presence or absence of genetic correlation between focal traits

- presence or absence of vertical transmission (i.e. non genetic factors transmitted from child to offspring)

As the entirety of the manuscript hinges on the validity of MR (which is known to break under AM), this is a critical missing piece of analysis.

Thank you for this comment, which is indeed a crucial one, on which the validity of the results relies. The crucial paper (cited by the papers mentioned by the reviewer) is by Hartwig et al.³¹. In that work various scenarios of assortative mating have been examined that may impact Mendelian randomization, but in situations where both exposure and outcome are measured in the same individual. We took this one step further and set up a model specific to causal effect estimations where exposures and outcomes are the same traits, but in different individuals (couples) to examine the potential issues. We have allowed for a more complex set of MR violations. We have updated the Methods section with a new subsection *Biases in the causal effect estimation*:

“Violations of the MR assumptions are different in applications like ours, where the exposure and outcome are the same trait, but measured in different individuals. Hartwig et al.³¹ explored the impact of assortative mating on MR, but for the classical setting of testing different traits in the same individual. Therefore, we set up a model (see Fig 8) specific to causal effect estimations where exposures and outcomes are the same traits, but in different individuals (couples) to examine the potential issues. We have allowed for a more complex set of MR violations, which we describe is below.”

Figure 8. Modelling the impact of parental effect and assortative mating on cross-sample MR. The model focusses on trait X in the offspring (index case) $[X_O^i]$ and that in the partner $[X_P^p]$. These traits are influenced by direct genetic effects $[G_O^i]$, direct (non-genetic) environmental effects $[E_O^i]$. Everything derived below works the same way even if G_O^i represents a single SNP (and hence E_O^i is heritable). The offspring environment is influenced by the parental environment $[E_P^i]$, the parental genes $[G_P^i]$ and the parental trait $[X_P^i]$. Parental characteristics (G, E, X) are simply defined as a rescaled average of the maternal and paternal traits, i.e. $X_P^i = (X_M^i + X_F^i) / \sqrt{2 \cdot (1 + \text{cor}(X_M^i, X_F^i))}$ in order to ensure that $\text{Var}(X_P^i) = 1$. Note that this simplification assumes that paternal and maternal effects are identical, which hold in general³². The direct causal effects of G_P^i, E_P^i and X_P^i on E_O^i are denoted by s_G, s_E and s_X , respectively. By definition, the correlation between mean parental genotype and offspring genotype is $1/\sqrt{2}$. Parental traits (E_P^i, X_P^i) cannot modify offspring genotype, thus they can influence X_O^i only via its environmental component (E_O^i). The same description holds for all corresponding variables of the partner. Finally, X_O^i

and X_0^p are paired such that predefined correlations between G , E and X are satisfied. For this, direct assortment coefficients r_G , r_E and r_X are defined between G , E and X , respectively.

As can be seen from Figure 8, parental effects induce a correlation of $\left(\frac{1}{\sqrt{2}}\right) \cdot (s_G + g \cdot s_X)$ between the offspring genotype (G_0^i) and the offspring environment (E_0^i). Also, one can note that the total couple correlation for X can arise from three independent sources: direct X -assortment [r_X], through direct assortment for E [$r_E \cdot e^2$] and through direct assortment through G [$r_G \cdot g^2$]. Thus, the expected correlation between X_0^i and X_0^p is $r_X + r_E \cdot e^2 + r_G \cdot g^2$. Similarly, the expected correlation between E_0^i and E_0^p is $r_E + r_X \cdot e^2 + r_G \cdot \left(\left(\frac{1}{\sqrt{2}}\right) \cdot (s_G + g \cdot s_X)\right)^2$ and between G_0^i and G_0^p is $r_G + r_X \cdot g^2 + r_E \cdot \left(\left(\frac{1}{\sqrt{2}}\right) \cdot (s_G + g \cdot s_X)\right)^2$.

Under this model, the expected effect of the index genotype on the index trait is

$$E[\hat{\beta}_{G^i \rightarrow X^i}] = g + \left(\frac{1}{\sqrt{2}}\right) \cdot (s_G + g \cdot s_X) \cdot e$$

while the expected effect of the index genotype on its partner's trait is

$$E[\hat{\beta}_{G^i \rightarrow X^p}] = \left(g + \left(\frac{1}{\sqrt{2}}\right) \cdot (s_G + g \cdot s_X) \cdot e\right) \cdot r_X + r_G \cdot g + \left(\frac{1}{\sqrt{2}}\right) \cdot (s_G + g \cdot s_X) \cdot r_E \cdot e$$

Therefore, the expectation of the estimated causal effect $X_0^i \rightarrow X_0^p$ can be written as

$$E\left[\alpha_{X_0^i \rightarrow X_0^p}\right] \approx \frac{E[\hat{\beta}_{G^i \rightarrow X^p}]}{E[\hat{\beta}_{G^i \rightarrow X^i}]} = r_X + \frac{r_G \cdot g + \left(\frac{1}{\sqrt{2}}\right) \cdot (s_G + g \cdot s_X) \cdot r_E \cdot e}{g + \left(\frac{1}{\sqrt{2}}\right) \cdot (s_G + g \cdot s_X) \cdot e}$$

Several parameters in this model can lead to the violation of MR assumptions.

Violation 1 ($s_G \neq 0$). This violation implies that parental genetics directly impact offspring environment (not through E or X). Such violation would most likely happen if G_P^i impacts another parental phenotype Y_P^i which has an impact on E_O^i , i.e. $s_G = g \cdot r_G(X, Y) \cdot s_Y$. Here $r_G(X, Y)$ refers to the genetic correlation between traits X and Y . The formula assumes that the effect of G_P^i to a secondary trait (Y) is expected to be its genetic effect on the primary trait (X) multiplied by their genetic correlation.

Violation 2 ($s_X \neq 0$). This violation is the classical parental rearing (or in other terms: dynasty) effect.

Violation 3 ($s_E \neq 0$). This violation does not lead to MR bias since there is no path from the offspring genotype to the offspring environment through the parental environment.

Violation 4. ($r_G \neq 0$). This violation allows direct assortment based on index and partner genetics, which most likely happen due to assortment for another trait (Y). Hence its typical size is $r_G = (g \cdot r_G(X, Y))^2 \cdot r_Y$. Here we made a similar assumption as for Violation 1. One (toy) example for such scenario could be educational attainment (X), whose genetic component is strongly associated with intelligence, for which additional assortment occurs.

Violation 5 ($r_E \neq 0$). This violation allows direct assortment based on the environment of the index and partner. A simplified example for this scenario is trait X being BMI and E representing “going to the gym”. In such case, beyond assortative mating for BMI, people are more likely to choose their partners from the gym (selecting for fitter individuals), hence the environment has an additional direct effect of the partner’s BMI.

To gauge the extent of the MR bias in realistic parameter settings, we have visualized the bias for a wide range of parameters. We set $r_X = 0.2$, the heritability to 20% (hence $g = \sqrt{0.2}$ and $e = \sqrt{1 - 0.2}$). Then two of the remaining four crucial parameters ($s_G, s_X; r_E, r_G$) were fixed and the other two were varied. For example, the chosen range for r_G assumed a value of $r_G(X, Y) = 0.3$ and r_Y range of $[-0.3, 0.3]$, using the abovementioned formula of $r_G = (g \cdot r_G(X, Y))^2 \cdot r_Y$. Similarly, we used the formula $s_G = g \cdot r_G(X, Y) \cdot s_Y$ to justify the explored range for s_G , with $r_G(X, Y) = 0.3$ and s_Y between -0.3 and 0.3 .”

Since we have derived an analytical formula for the causal effect estimate, there is no need to run simulations. To gauge the extent of the MR bias in realistic parameter settings, we have visualized the bias for a wide range of parameters and included these findings in the Results:

“In the Methods section we derived the analytical formula for the bias of cross-sample MR-estimates, which shows its exact dependence on each model parameter. Here we explored the extent of the bias under realistic ranges of model parameters and visualized it in Figure 6. First, we observed (Fig 6A) that r_G has little impact on the bias (since it is limited by the heritability and the genetic correlation) compared to r_E and they contribute additively. A similar relationship can be observed (Fig 6B) for s_G and s_X , the impact of the former is dwarfed by the latter for the same reason. Unsurprisingly, the largest bias emerges when both the direct environmental assortment (r_E) and parental effects (s_X) are the largest (Fig 6C). This can be complemented by direct environmental assortment (r_E) combined with parental genetic effects (s_G) (Fig 6D).

Figure 6. The impact of various model parameters on MR bias. We plotted the bias of the cross-sample MR estimates as a function of the proposed model parameters. Parameter r_G refers to direct genetic assortment; r_E : direct environmental assortment; s_G : direct parental genetic effect; s_X : direct parental trait effect. The different panels show the extent of bias when different pairs of parameters were co-varied: (A) r_G, r_E (B) s_G, s_X (C) r_E, s_X (D) s_G, r_E .

All in all, when direct environmental/genetic assortment is moderate, parental effects lead to negligible bias in the causal effect estimate between X_0^i and X_0^p . This is why we

focused our real data analysis efforts on identifying such X-associated (confounding) factors for which direct assortment may occur, so that r_E (and hence the MR bias) could be limited.”

The reviewer was particularly interested in two special cases of this framework:

1) The presence of genetic correlation between X_0^i and X_0^p . If the genetic correlation is only due to a causal effect between these traits, it does not introduce any bias. We, thus, guess that the reviewer was interested in the situation where $r_G \neq 0$, which does not seem to lead to excessive bias (Fig 5A).

2) The impact of non-genetic effects transmitted from parent to offspring, i.e. $s_E \neq 0$. It can be seen from the formula s_E does not induce an indirect genetic effect, hence does not have any impact on the MR analysis.

- The notion that the authors put forth prominently in their discussion, that women are choosers and men are courters is far beyond anything supported by their data. They observe no significant differences in causal effects across sexes after Bonferroni correction for 66 tests (which is liberal considering the total number of tests performed across the entire study). They then redo their hypothesis test choosing a more liberal threshold based (29 tests), and still don't finding anything significant. They select only the 15 nominally significant traits and do a t test on these traits only (which is virtually guaranteed to be significant due to the winner's curse) and get $p = .014$. If you keep altering your statistical procedure until you find something, the statistics lose meaning.

We agree that the sex-specific findings are borderline significant. Still, observing 15 out of 66 tests with sex-different $P < 0.05$ is an almost 5-fold enrichment ($P = 7.45 \times 10^{-8}$), which is a clearly significant evidence (even considering all tests performed in the entire paper) that the causal effects are not homogeneous between sexes. Second, we respectfully disagree with the reviewer, regarding Winner's curse for a one-sided t-test: just because we selected traits for which the difference was nominally significant, it does not lead to any bias in terms of female or male effects being larger. If the 15 selected traits had equal

male-female effects, for half of them we expect the male, the other half the female effect being larger, yielding no sex-bias. In such t-test the mean is expected to be zero, however, the variance will be larger than 1 due to the selection (but the t-test estimates this variance from the data, i.e. accounts for it). To further support our argument, we have performed (10,000) simulations where no sex-specific effect was imposed for 300 traits (to obtain 15 nominally significant traits [15/0.05]), male-vs-female effects were tested against each other and only those were selected that had nominally significant male-vs-female effects. We have seen that the average effect different for these selected traits is indeed zero and the t-test P-values (both two- or one-sided) follow a uniform distribution (see figure below), hence no Winner's curse bias exists.

* Additional concerns

- (line 54) The authors here suggest that assortative mating will lead concentration of genetic resources. This assertion is not supported by any the three papers they cite and is precisely the opposite of what they imply. Specifically, AM increases genetic variance, leading to dispersion, not concentration, of "genetic resources". It does not lead to a bimodal distribution. For every

pair of individuals at the extremes of the genetic spectrum mating assortatively, many more will be somewhere in the middle.

Indeed, the reviewer is correct. It was a typo, we meant to write “dispersion”, not “concentration”. This is now corrected in the text. With assortment, the homozygous proportions increase, leading to elevated genetic variance. We are not sure what the reviewer meant by “many more will be somewhere in the middle”: with (positive) assortment, not simply the extremes mate more probably, but those who are on the same side of the extreme (e.g. not very tall with very short), reducing heterozygotes. With negative assortment (opposites attracting each other), it is the reverse, heterozygote frequency goes up – but we have not seen example for such assortment for real traits.

- Every step of significance-based thresholds for which traits to consider in subsequent analyses induces winner's curse bias--I'd encourage the authors to analyze all traits. Also, perhaps one reason they didn't find much evidence for suppression effects is that they didn't examine confounding factors for traits without significant direct effects

We agree that reducing our analyses to the 64 traits with significant causal effects precludes the detection of confounder effect cancelling out direct effect. For this reason, we repeated this analysis for all 118 traits. As discussed above, filtering on traits with evidence for causal AM effect does not lead to bias for the sex- or age-differences, but it improves powers by reducing multiple testing. The analysis where we searched for confounding factors, was (naturally) reduced to the set of 43 traits, which showed significant difference between the phenotypic correlation and the causal effect estimates (otherwise there is no reason to look for potential confounders). All other analyses were run for all 118 traits. The Supplementary Tables have been extended to include all 118 traits and Tables 1-4 were merged so that all results can be found in the same place.

- I'd encourage the authors to rethink or supplement their use of the phrase "confounding" with alternative language or further discussion. Though it makes perfect sense in the context of causal analysis, it has the unfortunate connotation of implying that an observed association is spurious. This is relevant in light of recent evidence that simultaneous on multiple phenotypes can have more dramatic consequences for genetic architecture / inference. So even if a cross-mate association can be partly accounted for via an indirect path, this doesn't mitigate its impact on quantities of interest (and could even make them worse). I don't think the authors make this mistake, but their readers might.

The reviewer brings up a valid point: We refer to confounders as variables that induce correlation between two (other) variables, leading to differences in their causal effect and their correlation. For the cross-trait analysis, we have indeed incorrectly used the term confounder and we have now changed that sentence in the Methods to:

“We wanted to explore if one path was more dominant, in general, and if there was evidence for the presence of direct effects (or indirect effect with additional traits involved).”

and in the Discussion to:

“Of note, there were fewer cases in category three, where the causal effect from X_i to Y_p was not captured by γ or ρ , suggestive of either a direct effect X_i to Y_p or indirect effects through variables we have not explored.”

We have also admitted the limitation (Discussion) that we have not explored indirect causal paths for assortative mating effects:

“Third, we have not explored indirect causal paths from X_i to X_p through another variable (Z), measured in either individual. Given the limited evidence of direct cross-trait, cross-partner effects, the most likely such path would be $X_i \rightarrow Z_i \rightarrow Z_p \rightarrow X_p$, leading to an indirect effect of $\alpha_{X \rightarrow Z} \alpha_{Z \rightarrow X} \alpha_{Z_i \rightarrow Z_p}$. This implies a bidirectional causal effect between X and Z (within the same individual), but their product is expected to be negligible⁴⁴.”

This way we clearly distinguished indirect effects from confounded associations.

- The results about spousal convergence are not particularly compelling. First, time at shared address is a weak proxy for time spent together as many couples will have lived together prior to moving to their current residence (unless I'm misunderstanding?). Second, given the lack of age diversity in the UKB, there are few young couples. Third, the phenotypes for which spousal

convergence is likely the strongest (income) is measured at the couple level (household), not the individual level.

We have added these facts as severe limitations to address the question of couple convergence. In the Discussion:

“However, such discovery was massively hindered by the poor quality of the available proxy traits. To properly assess this question, longitudinal data including measures before couples were together would be best suited to disentangle the complex relationship between assortative mating and convergence.”

We fully agree with the review that the fact that the impact of partner similarity for a trait is not necessary weakened in case it is (partly) due to an indirect assortment. We did not mean to imply it anywhere in the text.

- the authors identified a larger number of spouses in the UKB that others have previously (e.g., Howe et al, Yengo et al [doi.org/10.1038/s41562-018-0476-3], Border et al [doi.org/10.1101/2022.03.21.485215]). It would be interesting to see which criteria are responsible for these differences.

The reviewer references three previous assortative mating publications which also used the UKBiobank data to analyze genetic relationships among couples and points to the discrepant sample size, specifically the larger sample size in our study.

In the Yengo et al. paper, the authors identified male-female couples 18,934 couples. Their criteria for couple identification was stricter, restricting to pairs with the same data among the following variables: “recruited from the same centre, reported living with their spouse or partner in the same type of accommodation, at the same location (east and north coordinates rounded to 1 kilometre), for the same amount of time, with the same number of people in the household, with the same household income, with the same number of smoker in the household, with the same Townsend deprivation index and with a genetic relationship of < 0.05 .” We did not match couples based on accommodation, east

or north coordinates, time in that location, number of people in household, household income, smokers in household or Townsend deprivation index. Although considering these variables increases confidence in the putative couples, we do not believe it is necessary that all these variables match among couples as there could be many reasons for discrepancy responses between couple pairs, for example differing interpretation or misunderstanding of the question, one couple moving in with the other (resulting in different amount of time in the location).

The Howe et al. paper identified 47,435 male-female couples (only ~4 thousand less than we identified). In this case, the criteria again limited to pairs who had been living at their address for the same length of time, which we did not exclude.

Finally, the Border et al. paper did not directly select putative couples as far as we can tell. However, they make reference to data from previous reports by Stulp et al. (<https://onlinelibrary.wiley.com/doi/10.1002/ajhb.22917>), which performed a literature review and meta-analysis of AM for human height, and Robinson et al. (<https://doi.org/10.1038/s41467-019-12424-x>), which used the interim UKBB release resulting in less than third of the full sample (only 152,736 participants) to start with which explains their much smaller number of couples (7,780 couples after filtering). Thus, this latter study is not comparable to ours.

We explain the differences in the updated Methods:

“Note that some studies have used more stringent criteria for couple definition. For example, Yengo et al.²² additionally requested couples to have been recruited in the same center, living in the same location, for the same amount of time, living with the same number of people with the same household income, same Townsend deprivation, same number of smokers, etc. Howe et al.²⁹ used a very similar definition to ours (resulting in 10% less couples than us), however restricted their analyses only to couples who reported to live at the same address for the same amount of time. We believe that discrepancies in self-reported data can occur frequently (due to misunderstanding, misreporting), hence decided to use a more liberal couple definition to increase sample size at the cost of minor misclassification.”

- (line 234) typo, i believe $\alpha_{z \rightarrow x}$ should be $\alpha_{y \rightarrow x}$, also a Z on line 273 should be a Y, unless I'm confused

Thank you for spotting both typos, we have corrected the sentence to:

“After identifying potential confounder traits, we combined these $\alpha_{y \rightarrow x}$ with the within couple causal effect ($\alpha_{y_i \rightarrow y_p}$), and calculated the correlation due to confounding as $C = \alpha_{y \rightarrow x}^2 \times \alpha_{y_i \rightarrow y_p}$ to determine the contribution each trait (Y) confounding the within couple correlation for trait X ($r_{x_i x_p}$). We subsequently calculated the ratio of the confounding estimate (C) and the correlation of X between partners as $C/r_{x_i x_p}$.”

and

“These confounding estimates were finally contrasted to the couple correlation values to explore the extent that each Y may confound couple correlations by examining the ratio between the two estimates (i.e. $C/cor(X_i, X_p)$).”

- (line 500) typo, "Figure 1C" should be 2C

Thank you, corrected.

* Additional questions

- Given that many confounders were identified for some traits, it seems likely that some of these confounders are themselves confounded. In (line 444), the authors note that they cannot simply add together the correlations across confounders, but it isn't clear why these effects cannot be

jointly estimated. It would be quite interesting to know if individual factors stand out (e.g., you could imagine SES accounting for a large amount of confounding signal observed for BMI/diet related traits) or if it's more of a wash (e.g., perhaps the confounding signals seen across metabolic phenotypes are largely overlapping).

This is a great comment and we have added this analysis to the Methods:

“We then explored those potential confounders, Y_k , with a significant impact on X ($p < 0.05/66$). As the confounding impact of each Y_k involves a within couple effect ($\alpha_{y_i \rightarrow y_p}$), as illustrated in Figure 7B we further filtered the remaining Y_k traits, to those with a significant within couple MR effect ($p < 0.05/[\text{number of remaining } Y_k]$). The couple correlation induced by confounder Y can be expressed as $\alpha_{y \rightarrow x}^2 \times \alpha_{y_i \rightarrow y_p}$. We restricted confounders to only those that had a correlation with X less than 0.8 in order to avoid using meaninglessly similar traits to X . Since potential confounders may be correlated, one could use MVMR to assess the joint contribution of these confounders on the couple correlation for trait X . This however led to numerical instability and we decided to rather prune confounders ($r^2 < 0.1$, prioritizing for larger $\hat{\alpha}_{y \rightarrow x}^2 \times \hat{\alpha}_{y_i \rightarrow y_p}$ values) and obtain $\hat{\alpha}_{y \rightarrow x}$ estimated from univariable MR, where IVs were pruned for independence ($r^2 < 0.001$). Finally, we plugged in the obtained causal effect estimates of (Y_1, Y_2, \dots, Y_m) on X , ($\hat{\alpha}_{Y_1 \rightarrow X}, \hat{\alpha}_{Y_2 \rightarrow X}, \dots, \hat{\alpha}_{Y_m \rightarrow X}$) into the estimator of the correlation induced by these confounders to get the estimate of total confounding $C = \sum_{j=1}^m \hat{\alpha}_{(Y_j)_i \rightarrow (Y_j)_p} \cdot (\hat{\alpha}_{Y_j \rightarrow X})^2$. The variance of such a sum was estimated as the sum of the variances of the individual terms (since they are uncorrelated). The variance of $(\hat{\alpha}_{Y_j \rightarrow X})^2$ was approximated by assuming $\hat{\alpha}_{Y_j \rightarrow X}$ following a normal distribution and used the general formula of $\text{Var}(X^2) = 4\mu^2\sigma^2 + 2\sigma^4$ for $X \sim N(\mu, \sigma^2)$. The variance of the product of $\hat{\alpha}_{(Y_j)_i \rightarrow (Y_j)_p}$ and $(\hat{\alpha}_{Y_j \rightarrow X})^2$ was estimated based on the formula for the variance of the product of independent random variables: $\text{Var}(XY) = \text{Var}(X)\text{Var}(Y) + \text{Var}(X)E^2(Y) + \text{Var}(Y)E^2(X)$.”

Furthermore, we updated the corresponding Result to:

“Finally, for each confounder we calculated the correlation due to confounding (C) as described above (see Methods and Figures 1B and 7B) and summed up these C values for all uncorrelated confounders. We then, for each trait, compared the difference in estimates (i.e. $\hat{r} - \hat{\alpha}$) to the estimated value C -sum (Figure 3B). One can observe some traits (e.g. systolic blood pressure) where the difference between partner correlation and causal effects can be well explained by the tested confounders, but for the majority of the traits, the observed confounders are not sufficient to account for the discrepancy (e.g. basophil count has strong positive confounders missing).

Figure 3: Phenotypic correlation in couples versus causal effects and evidence of confounder traits impacting the discrepant estimates. A. Scatter plot shows the within couple standardized MR-estimates ($\alpha_{x_i \rightarrow x_p}$) versus the phenotypic correlation among

couples ($r_{x_i x_p}$); error bars represent 95% CIs. A two-tailed z-test was used to test for a significant difference between the estimates. After adjusting for the number of effective tests ($p < 0.05/66$), 43 significant differences were identified (shown in dark blue), where 3 traits showed larger MR-estimates compared to correlation, and 40 traits showed larger correlation compared to MR-estimates. The identity line is shown in black. Labelled pairs are discussed in the main text. Abbreviations: SBP: systolic blood pressure; FVC: forced vital capacity; NC: North coordinate. B. Scatter plot shows the difference in phenotypic correlation and MR-estimate versus the C-sum value (estimating the correlation induced by measured (uncorrelated) confounders) for each trait where the phenotypic correlation was greater than the MR-estimate (number of traits = 39); error bars represent 95% CIs. The identity line is shown in black. Abbreviations: SBP: systolic blood pressure; FVC: forced vital capacity; NC: North coordinate.”

We also updated Supplementary Table 2 with the C-sum values.

Reviewer #2:

Remarks to the Author:

Sjaarda and Kutalik present very interesting research using genetic and phenotype data to tease apart why spouses are similar. I think this is an important research question as spouses are correlated for nearly every phenotype and often this does not relate to mate-choice on the index phenotype but rather to assortment on other phenotypes such as educational attainment. For example, spouses are correlated for many biomarkers which as non-visible phenotypes are very unlikely to influence mate choice. The results presented here are very compelling evidence that partner similarities are driven by a smaller set of phenotypes.

I had a few concerns about the language and some of the conclusions detailed below. I am confident that the manuscript will be an important contribution to the field once refined.

We thank the reviewer for their kind comment.

1) Semantics of assortative mating.

Title “The contribution of mate-choice, couple convergence and confounding to assortative mating”

Abstract “Increased phenotypic similarity between partners, termed assortative mating (AM), has been observed for many traits.”

Introduction “In human populations, phenotypic similarity exists between partners compared to random pairs, a phenomenon known as (positive) assortative mating (AM).”

I personally think that equating ‘spousal phenotypic similarities’ and ‘assortative mating’ as synonymous is confusing. To me, assortative mating refers to individuals preferring to select (and perhaps remain in a relationship with) a phenotypically (dis)similar partner (which the authors have referred to as “mate choice”). I think it reads oddly when assortative mating is defined as including post-mating mechanisms such as spousal convergence from partner interaction effects or the shared environment. For example, if a cohabiting spouse-pair are both exposed to asbestos in their house and then develop lung cancer, would this be included under assortative mating?

I much prefer the way the authors have explained the different mechanisms in Figure 1 (below) where assortative mating (defined as mate choice on phenotypes, geography and social factors etc) is distinguished from post-mating mechanisms.

Illustrates a trait (given by the colour blue) which is under assortative mating, either directly (through mate choice) or due to confounding factors such as shared geography, cultural or religious status or socioeconomic measures. Subsequently, this trait may also undergo post-mating convergence which could be due to direct causal influence from one partner on the other (i.e. through imitation or influence) or due to confounding factors such as shared environment.

The authors are not alone in equating assortative mating and phenotypic similarities and so I’m not necessarily requesting that they necessarily change their terminology. I’m curious what the authors think is the most appropriate terminology and whether they think AM and spousal phenotypic similarities are synonymous. I think it would also be good to be consistent with

language throughout as Figure 1 seems to define things slightly differently to the introduction and the abstract.

I would personally be inclined to define AM as all mechanisms inducing spousal similarities at pairing, including direct assortment, indirect assortment and assortment on confounders such as geography. Then define the post-mating mechanisms of couple convergence and shared environmental effects separately. Do the authors think this would be appropriate?

We agree with the view of the reviewer, it makes more sense. We have edited the manuscript to reflect this distinction. Systematically we referred to the global phenomenon as “partner similarity” / ”phenotypic similarity between partners” / “PS” / “couple correlation” and we distinguish it from “assortative mating” / “assortment” / “mate choice” throughout the manuscript. On some occasions (when talking about cross-trait assortment), we use the term “assortative mating” in the more general meaning, to simplify some sentences.

2) Direct assortment - Lines 56 to 66.

Direct assortment here would include both mate choice effects and then also partner interaction and shared environmental effects? For example, if BMI does not influence mate choice but couple’s BMI converges over time then this would be direct assortment? Perhaps worth expanding on this in the text as I got a bit confused going from the “three underlying phenomena” to direct and indirect assortment.

According to our definition, the example of the reviewer falls into category (iii), post-mating convergence. Indeed, we could have split up that category into direct and indirect convergence. We have now done this to improve clarity.

“[...] additionally modified by post-mating convergence (during cohabitation), which could be further divided into direct and indirect convergence (depending whether the convergence happens for the focal trait or for a correlate).”

3) Spouse definitions

In the methods it states that spouses were defined using household data, but it doesn't explain how these households were defined. Please add in extra detail including the data fields used to assign study participants to the same household, e.g. was this based on shared home coordinates? "Within this sample, we retained individuals coming from households with exactly two unrelated, opposite-sex individuals, leaving 108, 898 participants."

The household ID for each individual was sent to us separately upon request and it is not part of the classically available fields of the UK Biobank. We specified this in the Data availability section:

"The UK Biobank data is available upon an application procedure (<https://www.ukbiobank.ac.uk/enable-your-research/apply-for-access>). The household information was separately requested and not part of the variables listed on the website (<https://biobank.ndph.ox.ac.uk/showcase/index.cgi>)."

Could you please also return the derived spouse data to UKB, I believe this is mandatory so that other researchers can use them. Perhaps add this in data availability?

The derived spouse information cannot be easily fed back, since the IDs are application specific. We extended the Data availability section (see above).

4) Proxy for relationship length

The authors use length of time at current address as a proxy for relationship length. I'm concerned that this may have limitations as I'm sure many couples do move around. The UKB generation had a relatively easy time on the housing market compared to the generation of today and so many would have bought increasingly larger properties over time.

Could the authors show the distribution of the proxy variable? Does the distribution seem feasible as a proxy for relationship length? The authors do discuss this as a limited proxy.

We agree that such measure is a poor proxy, but no better one is available (we specifically ask UKB about this). We have extended the description of this weakness in the updated Discussion:

“Second, with the current data, we were not able to find strong evidence for couple convergence over time. We did make use of both age and time-together data (proxied by time at the same address) to help shed light on this question, and were able to show that certain traits indeed appear to converge as a couple spends more time together while other traits appear to be more important in the selection process (i.e. true assortative mating). However, such discovery was massively hindered by the poor quality of the available proxy traits. To properly assess this question, longitudinal data including measures before couples were together would be best suited to disentangle the complex relationship between assortative mating and convergence.”

5) “Our results point to AM being stronger among females compared to males which is consistent with the notion first put forth by Darwin, that females are, in general, choosers, while males are courters”.

Is an alternative explanation that women have stronger effects on their partner in the relationship rather than a mechanism relating to partner selection? MR of female genotype to male phenotype should pick up the effects of the woman's genotype on their spouse's phenotype. For example, if the woman is lactose intolerant (LCT genotype) and then buys less milk, this would affect the milk consumption of her partner.

We fully agree with the reviewer and we have modified the corresponding section to:

“Our results point to female-to-male effects being on average stronger than male-to-female effects. This difference may reflect mate choice or directionality of couple convergence. The first possibility would be consistent with the notion first put forth by Darwin, that females are, in general, choosers, while males are courters³¹. While the contribution of the latter phenomenon would imply that couples converge phenotypically closer to the female partner.”

6) Measurement error

Phenotypes can be susceptible to measurement error while genotypes (if imputed well) are often measured accurately. How could phenotypic measurement error potentially affect the different comparisons in the manuscript, specifically the comparison between MR estimates and phenotypic correlations?

Simulations would be helpful to understand if measurement error could have affected any of the conclusions. At the very least, I think measurement error should be acknowledged as a limitation.

This is an excellent point we have ignored in the original submission. Indeed, measurement noise in the phenotype can reduce the observational correlation substantially (noise with a variance $s\%$ of that of the trait leads to $1/(1+s/100)$ deflation of the correlation). Such noise, however, has negligible impact on the causal effect estimation, since the regression dilution bias equally reduces the effect on the exposure and the outcome. This observation is very specific to testing the same phenotype in different individuals, since in this situation, the measurement noise is (by definition) equal for the exposure and the outcome. We have added this comment to the Discussion:

“Of note that phenotypic correlations in couples are impacted differently by measurement noise than causal effect estimates. While the former estimates are attenuated by a factor of $\frac{\text{Var}(y)}{\text{Var}(y)+s^2}$ in case the true phenotype y is measured with a noise with variance s^2 , the causal effect estimates do not change noticeably because the exposure and outcome effects are equally diluted. This could lead to an underestimation of confounding effects in our results and may explain why for three traits we observed larger causal effect than correlation.”

Unfortunately, without the knowledge of the extent of measurement noise we cannot correct for this issue. Still our confounding estimates are valid as a lower bound.

Minor comments

1) Abstract line 15 – unclear to me what “causal relationship between partners” means, could you redraft using the assortment terminology?

We corrected to:

“Sixty-four of the tested traits showed evidence for a direct assortment/convergence between partners [...]”

2) Introduction line 47 – would we use an individual’s genome to predict their partner’s phenotypes? Maybe rephrase to “is correlated with”?

“As a consequence of partner similarity, the genome of an individual correlates with certain traits of their partner.”

3) Intro line 54 – “ultimately resulting in a concentration of (genetic) resources” isn’t very clear what this means, please elaborate.

We clarified it:

“For instance, increased phenotypic similarity could naturally imply genetic similarity, leading to variants that are otherwise independent to become correlated, and ultimately resulting in a dispersion of (genetic) resources, i.e. leading to an increased proportion of homozygous individuals.”

4) Line 130 – “The random distribution of genetic variants at birth reduces the possibility of confounding or reverse causation” Germline genotype being fixed from birth relates to reverse-causation being unlikely, suggest rephrasing.

Yes, being germline excludes reverse causation, but the fact that it is randomly distributed (i.e. mimicking a randomized control trial) suggests that it is also less prone to confounding.

5) Is it worth also including spousal genetic correlations for comparison? Genetic correlations cannot be impacted by post-mating mechanisms so can only reflect mate choice.

Indeed, genetic correlations may not reflect post-mating effects. However, ascertainment

bias might nuance it, since the study is biased for couples that stayed together longer, which may have a genetic basis increasing genetic correlation reflecting post-mating effects. The reason why we intentionally avoided genetic correlations is that those still reflect indirect assortment, much more so than Mendelian randomization. The imperfectly proxied time spent together analysis indicates that post-mating convergence is relatively negligible compared to indirect assortment, since they can still reflect the impact of heritable confounders. For example, the genetic correlation of BMI between partners may involve shared diet and education, both of which have genetic components (<https://pubmed.ncbi.nlm.nih.gov/35653391/>, <https://pubmed.ncbi.nlm.nih.gov/35361970/>). The most comprehensive study (<https://www.nature.com/articles/s41562-016-0016>) exploring genetic and phenotypic correlations between partners revealed that for most traits these two quantities closely agree. The exception to this was BMI, where phenotypic correlation was generally larger than the genetic one, possibly reflecting post-mating convergence. However, the other discrepant trait was educational attainment, where the genetic correlation was markedly higher, possibly due to the fact that the genetic component of educational attainment correlates better with traits (e.g. cognitive performance) which that are under stronger assortment. This highlights that genetic correlations – while less prone to reflect post-mating convergence – may be more sensitive to indirect effects. We have briefly included this in the Discussion:

“A complementary approach could be to contrast genetic and phenotypic correlations; however it is hard to tell whether differences reflect post-mating effects or different (genetic and environmental) correlation to traits under primary assortment.”

Reviewer #3:

Remarks to the Author:

In this paper, the authors use mendelian randomization to test for causality in assortative mating for 118 traits in a UKB sample of ~52k couples. They find that about half of these traits produced evidence consistent with a causal relationship between partners, and that income represented the largest confounder in cross-partner correlations. They additionally find little evidence for couples growing more similar over time.

This is a nice paper, with a comprehensive set of phenotypes and a big UKB sample. The authors find some interesting effects that align well with my own understanding of existing

literature in this area. In sum, this research fills an important gap in our understanding of the context and underlying mechanisms of assortative mating, and I recommend it for publication in NHB. Relatively minor editorial suggestions for the authors follow.

Introduction

In your introductory coverage of various phenotypes that we know to represent AM, I would recommend including religious and political attitudes/values—these in fact represent some of the very strongest AM traits that we know of. See for example a recent adoption study by Willoughby, Giannelis, Ludeke, Klemmensen, Norgaard, Iacono, Lee and McGue (2021) which found enormous spousal correlations for political orientation and related phenotypes as well as religiousness. Earlier studies looking at AM for social attitudes include Kandler, Bleidorn & Riemann (2012) and Eaves et al. (1999).

Thank you for pointing out these important traits, which we have now added to the Introduction.

“This has been observed across a wide variety of traits, including anthropometric measures (such as BMI and height), socioeconomic factors, various behavioural (religious views¹, social attitudes²) and lifestyle measures, (including diet, smoking habits, hobbies, among others), and even disease risk³⁻¹⁰. “

Apropos of the above studies, it might also be worth briefly discussing an implication of strong assortative mating for some phenotypes: that AM directly influences ACE estimates, which may lead to (e.g.) increasing or decreasing religious and political polarization over time.

Laurence Howe and colleagues also had a paper recently (2021) looking at genetic effects of AM in the UKB, perhaps worth citing in the context of the introduction and subsequent discussion of AM at the genomic level.

Thank you for pointing out these missing pieces of literature. We have now added this paper and two others that raise various consequences of AM:

“Another consequence of partner similarity is that it impacts genetic association studies. For example, it has been shown that PS leads to inflation in both heritability²³ and

genetic correlation estimates²⁴ even when estimated from ACE models²⁵. Moreover, it can introduce collider bias in within-spouse association models²⁶

Lines 82-83: "(whereas classical MR designs involve a single individual, e.g. BMI risk on CAD)." The acronym "CAD" is not explained in the text and might not be known to readers unfamiliar with medical phenotypes.

Thank you, we spelt out the acronym (especially since this is not used again in the paper).

Overall I think there are too many parentheticals in the introduction—its last sentence for example (lines 91-93) has two that are nested.

We have removed the inner parenthesis from the mentioned example. We have also removed another instance of parenthesis from the Introduction. We found that sometimes these parentheses are useful for concise sentences.

Methods

Some of the coding-related details might be better suited to the supplement. Lines 116-118, for example, would read much better if translated to plain English (e.g., "individuals exhibiting aneuploidy were not included in the final sample"). It's also unclear what "excess relatives" means in that variable name.

We moved the detailed description to the legend of Supplementary Figure 1. We also explained the meaning of "excess relatives" in the figure caption.

Lines 267-268: How/why were these traits chosen for potential cofound variables? I assume from supp table 1 that it's because these traits are all in the top 10 for AM, but then why not religious group? It would be good to see in the main text if there's some other a priori reason for these choices.

We would have liked to include religion as confounder; however, it is not available in the UK Biobank (to the best of our knowledge). The decision was somewhat ad hoc, based on a manual screening of lifestyle/geography variables. We now admit the arbitrariness in the manuscript:

“The choice of these traits was somewhat ad hoc, mostly driven by previous evidence for driving trait similarities and themselves showing strong couple correlation in the UK Biobank.”

Lines 283-284: Using “length of time at current address” as a direct proxy for length of relationship seems potentially problematic: This would not account for those couples wherein one partner moves in with another at their existing address. What was done in cases where the “length of time at current address” differed between two partners in a couple?

When partners reported different time at the current address, we used the minimum of the two values, reflecting a situation when one person move in to the flat of the partner. Indeed, this is a poor proxy, but we had no better variable (which was confirmed by UK Biobank via personal communication). We have a more extensive acknowledgements of this shortcoming in the Discussion:

“We did make use of both age and time-together data (proxied by time at the same address) to help shed light on this question, and were able to show that certain traits indeed appear to converge as a couple spends more time together while other traits appear to be more important in the mate selection process (i.e. true assortative mating). However, such discovery was massively hindered by the poor quality of the available proxy traits. To properly assess this question, longitudinal data including measures before couples were together would be best suited to disentangle the complex relationship between assortative mating and convergence. A complementary approach could be to contrast genetic and phenotypic correlations; however it is hard to tell whether differences reflect post-mating effects or different (genetic and environmental) correlation to traits under primary assortment²⁰.”

Results

It would be interesting to see the results broken down by trait category. Most of the traits studied fall into one of at least three clearly-defined categories, anthropometric traits (BMI, metabolism, appendage fat mass), behavioral traits (education, income, video games, etc), and lifestyle traits (smoking, alcohol, and so on). This intuitive grouping is also referenced in the discussion, with a claim that these groups differed by statistical power (lines 627-630), but readers can't easily confirm that this is true without any actual grouping variable to reference.

We have generated a new Supplementary Table 1 with all 118 variables, including both their phenotypic correlation, sample size, causal effects, confounding factors, sex-specific effects, age-binned results, etc. We also merged Supplementary Tables 1-4 to have all information in one place and added a new column "category" and grouped the traits by category. The table caption reads as follows:

"Supplementary Table 1: Summary of all results for all 118 traits including number of pairs (n_pairs), phenotypic correlation among couples ($pheno_couple_r2$, $pheno_couple_r$, $pheno_couple_r_se$), MR-estimates ($n_exposure$, $n_outcome$, IVW_beta , IVW_se , IVW_pval), differences between phenotypic estimates and MR estimates ($couple_r_vs_IVW_diff_pval$), sex-specific MR estimates and differences (sex_het_pval , IVW_beta_male , IVW_se_male , IVW_pval_male , and $female$), and model estimates to assess the trend across bins by both age and time-spent-together, for (1) MR-estimates (weighted by SE), (2) Pearson correlations and (3) Spearman correlations (column names are given as bin_slope_beta , bin_slope_se , bin_slope_pval , with suffixes MR_wt , $pearson$, and $spearman$ to denote the different models, and age and $time_together$ to denote the different binning category). Note that n_pairs refers to the number of couples that have each have data for the specified phenotype to compute couple correlations ($couple_r2$, $couple_r$, etc), $n_exposure$ refers to the number of participants in the Neale models (sample size summed across male and female models), and $n_outcome$ refers to the number of participants in the models estimating the effect of SNPs on the outcome of interest. Specifically, the model estimated the association between each genetic instrument measured in the index individual with the phenotype measured in the partner using the UKBB partner data set described. As these models were performed in males and females separately and do not require phenotype information for each partner, the result $n_outcome$ is larger than n_pairs ."

Supplement

My only major comment for the supplement is to recommend including a table that includes all 118 phenotypes and their IVW statistics, rather than just the causal-significant ones. This is something readers will definitely be interested in and it would be especially useful for researchers to have a singular reference for all 118 traits.

It would also be good for replication and etc purposes to see the sample size for each trait in supp table 1.

This is a great comment and indeed, such a table is very useful for replication purposes and it also allowed us to merge many different supplementary tables into a single comprehensive one. We also included the sample size for each trait.

Minor picky comments:

It's really interesting to go through the IVW values for all of the sig traits (supp table 1). But I do have a few questions about some of the trait filtering decisions. In the methods it's stated that redundant phenotypes were removed. It's not clear to me how "(home and birth) location (north coordinate) and "location (east coordinate)" are not redundant—would it not make sense to use one measure of "home location" and one for "birth location"? How should the reader interpret the fact that AM for "north coordinate" is higher (eyeballing it, probably significantly higher) than for "east coordinate"? Does this correspond to the fact that the UK is longer than it is wide, so more variability north-to-south?

We defined redundancy in the paper as follows:

"Specifically, (i) left-side body traits (highly correlated with right-side) were removed, (ii) we retained only one of the duplicated phenotypes for BMI and weight (retaining UKBB data fields 21001 and 21002, respectively), and (iii) all "qualifications" data was removed (corresponding to UKBB field 6138) due to the availability of finer-scale correlated variables, such as "age completed full time education" (data field 845)."

We decided to keep both home and birth locations because they capture mobility/migration patterns, which may be particularly meaningful (as shown here:

<https://pubmed.ncbi.nlm.nih.gov/31636407/>). We now admit that this removal was not a systematic process:

“This ad hoc procedure was meant to capture only major redundancies.”

The question comparing north-south vs east-west assortment is intriguing. In terms of actual distance, couples differ on average by 76km in their North-South (NS) coordinate, but only by 52km in their East-West (EW) coordinate. This means that EW-wise couples are more similar in terms of km distance, but of course any random pair is also more similar in terms of EW location, so it is not a good measure of couple similarity. The average difference [more precisely the square root of the mean squared difference] is proportional to $\sqrt{2 * (1 - r) * Var(y)}$, where r is the correlation and $Var(y)$ is the variance of that phenotype. Since the NS variance is ~3.2 times larger than the EW variance and the couple correlation of NS was ~0.59, while for EW was ~0.35, we expect a ~42% larger NS distance (which we observed). Thus, the reviewer is correct, the larger the trait variance, the larger correlation is expected for the same phenotypic difference on the original scale. We believe that correlation is a more meaningful measure of trait similarity. However, we can only speculate whether stronger NS-similarity is due to stronger cultural/geographic barriers NS than EW.

Is there any indicator for handedness in the UKB? Reducing the phenotypes to right-body traits makes sense, but I do wonder if handedness would make a difference. AM for arm fat mass (e.g.) might look different if traits were collapsed to “arm fat mass (side corresponding to handedness)”.

This is an interesting point we have not previously explored. As it turns out, handedness has very minor impact on the couple phenotype correlation values, for two reasons: i) only ~90% of UKB participants are right-handed; (ii) handedness has small impact on arm fat. For arm fat, for example, we observed a 2-3% relative increase in the correlation when restricting the analysis to right-handed couples only (moving from 0.141 to 0.144).

Decision Letter, first revision:

5th October 2022

Dear Professor Kotalik,

Thank you once again for your revised manuscript, entitled "The contribution of mate-choice, couple convergence and confounding to phenotypic partner similarity," and for your patience during the re-review process.

Your manuscript has now been evaluated by Reviewers 1 and 3 from the original round of review. All reviewer feedback is included at the end of this letter.

Although one of the reviewers found your manuscript to have improved during revision, Reviewer #1 points to important remaining issues with how you interpret your statistical evidence and how they support or don't your claims on sex differences. We remain interested in the possibility of publishing your study in Nature Human Behaviour, but would like to consider your response to these outstanding concerns in the form of a revised manuscript before we make a decision on publication.

In your revision, please address all concerns raised by Reviewer 1#, including potential errors and issues with trait non-independence. Please remove all interpretations of nonsignificant results in the context of sex differences.

In sum, we invite you to revise your manuscript taking into account all reviewer and editor comments. We are committed to providing a fair and constructive peer-review process. Do not hesitate to contact us if there are specific requests from the reviewers that you believe are technically impossible or unlikely to yield a meaningful outcome.

We hope to receive your revised manuscript within 4 weeks. I would be grateful if you could contact us as soon as possible if you foresee difficulties with meeting this target resubmission date.

- Include a "Response to the editors and reviewers" document detailing, point-by-point, how you addressed each editor and referee comment. If no action was taken to address a point, you must provide a compelling argument. This response will be used by the editors and reviewers to evaluate your revision.
- Highlight all changes made to your manuscript or provide us with a version that tracks changes.

[REDACTED]

We look forward to seeing the revised manuscript and thank you for the opportunity to review your work. Please do not hesitate to contact me if you have any questions or would like to discuss these revisions further.

Sincerely,

Arunas Radzvilavicius, PhD
Editor, Nature Human Behaviour
Nature Research

REVIEWER COMMENTS:

Reviewer #1:

Remarks to the Author:

The authors continue to claim that their results support the notation "that females are, in general, choosers, while males are courters." This is far beyond their data and their rebuttal of my concerns contains multiple errors. Further, in examining their data directly, I am even less convinced by their interpretation.

First, they take 64 traits with significant causal effects in partners after correction for multiple testing. Differences. None of these were significant for sex-differences in these effects after correcting for the effective number of tests based on the spectrum of the covariance matrix. This is all they have to report. However, they go on to say "15 traits showed a nominally significant difference between sexes, which is four times higher than expected ($p_{\text{binomial}} = 7.45 \times 10^{-8}$)". Then, they do a paired t-test of sex-specific effects for these 15 traits (which were chosen on the basis on significance), and report that p-value.

First, as a minor point, this pbinomial-value is wrong (see, e.g., in R: `binom.test(15,64,.05)`), the correct value is an order of magnitude larger. Second, they are reporting a p-value for differences in effect estimates, after selecting the traits on the basis of significance of a parameter reflecting sex differences. This p-value will not be calibrated. In their response to my comment about this in the first round of revisions they present a simulation I was unable to replicate. I observe a type-1 error rate of ~15% rather than the supposed 5% when doing something analogous:

```
#####  
set.seed(2022)  
mres <-
```

```

replicate(1e3, simplify=F, {
  results <- replicate(66, simplify = T, {
    males <- rnorm(1000)
    females <- rnorm(1000)
    tt <- t.test(males, females) ## het test
    c(paired_diff=mean(males-females), pval=tt$p.value)
  })
try(t.test(results['paired_diff'],results['pval'],<.05])) ## paired ttest
})

## type 1 error rate for paired t-test setting the pvals to one
## for simulations where no traits were signifcant.
null_rejects <-
table(sapply(mres,
  functon(x) ifelse(class(x) == 'try-error',1, x$p.val))<.05)
null_rejects / sum(null_rejects)
#####

```

Considering the p-value they report is borderline, this is non-negligible, and poor practice in general.

Third, let's look at these 15 traits (pulled from the supplementary Excel doc):

1. "Job involves mainly walking or standing",
2. "Job involves heavy manual or physical work",
3. "Lifetime number of sexual partners",
4. "Forced expiratory volume in 1-second (FEV1), Best measure",
5. "Basal metabolic rate",
6. "Leg fat mass (right)",
7. "Whole body fat mass",
8. "Weight",
9. "Body mass index (BMI)",
10. "Trunk fat mass",
11. "Trunk predicted mass",
12. "Trunk fat-free mass",
13. "Arm fat-free mass (right)",
14. "Arm predicted mass (right)",
15. "Whole body water mass"

The way I have grouped these 15 traits is not an accident, there are at most 4-5 different constructs here, and these are mostly traits with big sex differences in variability. The number 15 they're basing the (incorrect) binomial test on is a fiction, these are not independent traits. See attached figure for how the data look for all 64 traits, highlighting the "15" traits selected post hoc on the basis of significance.

I did not look at the supplementary data the first time around, but this is a major interpretive caveat and the fact that the authors either didn't notice this or glossed over this is unnerving. This is certainly not enough to continue to claim that "men are courters and women are choosers". Instead, they observe that for a few traits, most of which are either BMI/weight or working as a laborer, mate preferences can't be constrained to be equal. Is this preference or differences in what these traits mean socioeconomically and how they vary across sexes. I cannot recommend that this work be published unless these claims are removed and the fact that many of these traits are essentially the same trait is discussed at length.

The paper is interesting enough without this, the question of whether cross-mate similarity reflects preference or convergence is worth asking. They have done a good job responding to most of my other concerns. However, the more time I have spent diving into their results, the more concerned I have become.

Reviewer #3:

Remarks to the Author:

The authors have adequately addressed my concerns, and I believe the manuscript to be greatly

improved as a result of incorporating reviewer 1 and 2's suggestions. I believe the paper to be suitable for publication. The figures are excellent!

Author Rebuttal, first revision:

We would like to thank Reviewer #1 for the additional insights and comments, which we addressed below.

Reviewer #1:

Remarks to the Author:

The authors continue to claim that their results support the notation “that females are, in general, choosers, while males are courters.” This is far beyond their data and their rebuttal of my concerns contains multiple errors. Further, in examining their data directly, I am even less convinced by their interpretation.

First, they take 64 traits with significant causal effects in partners after correction for multiple testing. Differences. None of these were significant for sex-differences in these effects after correcting for the effective number of tests based on the spectrum of the covariance matrix. This is all they have to report.

However, they go on to say “15 traits showed a nominally significant difference between sexes, which is four times higher than expected ($p_{\text{binomial}} = 7.45 \times 10^{-8}$)”. Then, they do a paired t-test of sex-specific effects for these 15 traits (which were chosen on the basis on significance), and report that p-value.

First, as a minor point, this p_{binomial} -value is wrong (see, e.g., in R: `binom.test(15,64,.05)`), the correct value is an order of magnitude larger.

Indeed, the reviewer is correct. We incorrectly applied the `pbinom()` function and computed the probability to observe more than 15 nominally significant P-values instead of at least 15. The P-value should have been $4.687767e-07$ for this binomial test.

Second, they are reporting a p-value for differences in effect estimates, after selecting the traits on the basis of significance of a parameter reflecting sex differences. This p-value will not be calibrated. In their response to my comment about this in the first round of revisions they present a simulation I was unable to replicate. I observe a type-1 error rate of ~15% rather than the supposed 5% when doing something analogous:

```
#####
set.seed(2022)
mres <-
replicate(1e3, simplify=F, {
  results <- replicate(66, simplify = T, {
    males <- rnorm(1000)
    females <- rnorm(1000)
    tt <- t.test(males, females) ## het test
    c(paired_diff=mean(males-females), pval=tt$p.value)
  })
try(t.test(results['paired_diff',results['pval'],]<.05)) ## paired ttest
})

## type 1 error rate for paired t-test setting the pvals to one
## for simulations where no traits were significant.
null_rejects <-
table(sapply(mres,
function(x) ifelse(class(x) == 'try-error',1, x$p.val))<.05)
null_rejects / sum(null_rejects)
#####
```

The only difference between our and the reviewer’s implementation of the null simulation is that we used 300 trials (and not 66) in order to get 15 nominally rejected as we observed for real data. If we change the reviewer’s code replacing “66” with “300”, it gives the same result as we suggested:

FALSE TRUE
0.948 0.052

The reason for this difference (between 300 and 66) is due to the violation of the t-test’s normality assumption: When there are only 66 trial runs, on many occasions we will have

low number of observations selected at $P < 0.05$ (on average $3.3 = 66 \cdot 0.05$) and the difference statistics are, by definition, truncated Gaussians, $Z = (X \mid |X| > t)$. Therefore, the t-test P-values are badly calibrated. Hence, we replaced the "t.test" with "wilcox.test" and kept the 66 tests just like the reviewer did. The resulting P-values are rather conservative than lenient:

FALSE TRUE

0.988 0.012

When we apply Wilcoxon ranksum test to the sex-difference value for the 15 traits with nominally significant sex-difference P-values, it yields $P = 0.002625$. Hence, this is not the source of the problem, but rather the dependence between the tests as spotted by the reviewer, see below.

Considering the p-value they report is borderline, this is non-negligible, and poor practice in general.

Third, let's look at these 15 traits (pulled from the supplementary Excel doc):

1. "Job involves mainly walking or standing",
2. "Job involves heavy manual or physical work",
3. "Lifetime number of sexual partners",
4. "Forced expiratory volume in 1-second (FEV1), Best measure",
5. "Basal metabolic rate",
6. "Leg fat mass (right)",
7. "Whole body fat mass",
8. "Weight",
9. "Body mass index (BMI)",
10. "Trunk fat mass",
11. "Trunk predicted mass",
12. "Trunk fat-free mass",
13. "Arm fat-free mass (right)",
14. "Arm predicted mass (right)",
15. "Whole body water mass"

The way I have grouped these 15 traits is not an accident, there are at most 4-5 different constructs here, and these are mostly traits with big sex differences in variability. The number 15 they're basing the (incorrect) binomial test on is a fiction, these are not independent traits. See attached figure for how the data look for all 64 traits, highlighting the "15" traits selected post hoc on the basis of significance.

We thank the review for this important observation we missed. This is crucial and indeed it invalidates the claim about female effect being stronger. For this reason, we have removed all claims about sex-specificity in the abstract, the Discussion and changed the Results to

"We assessed the 64 significant results for significant sex-differences, but did not identify any after adjusting for the effective number of tests among the remaining traits based on their pair-wise correlation matrix ($p < 0.05/29$)."

Due to the correlation between the 64 traits, the binominal test is not valid. While the enrichment ratio of 4.7 is valid, but its corresponding P-value would not be significant accounting for the dependence (~4 out of 29 effective traits). Therefore, we removed all mentions of sex-differential effects.

I did not look at the supplementary data the first time around, but this is a major interpretive caveat and the fact that the authors either didn't notice this or glossed over this is unnerving. This is certainly not enough to continue to claim that "men are courters and women are choosers". Instead, they observe that for a few traits, most of which are either BMI/weight or working as a laborer, mate preferences can't be constrained to be equal. Is this preference or differences in what these traits mean socioeconomically and how they vary across sexes. I cannot recommend that this work be published unless these claims are removed and the fact that many of these traits are essentially the same trait is discussed at length.

We agree with the reviewer that such statement was not supported by the data and have removed all such claims.

Decision Letter, second revision:

25th October 2022

Dear Dr. Kutalik,

Thank you for submitting your revised manuscript "The contribution of mate-choice, couple convergence and confounding to phenotypic partner similarity" (NATHUMBEHAV-22041009B). It has now been seen by the original referees and their comments are below. As you can see, the reviewers find that the paper has improved in revision. We will therefore be happy in principle to publish it in *Nature Human Behaviour*, pending minor revisions to satisfy the referees' final requests and to comply with our editorial and formatting guidelines.

We are now performing detailed checks on your paper and will send you a checklist detailing our editorial and formatting requirements within a week. Please do not upload the final materials and make any revisions until you receive this additional information from us.

Sincerely,

Arunas Radzvilavicius, PhD
Editor, *Nature Human Behaviour*
Nature Research

Final Decision Letter:

Dear Professor Kutalik,

We are pleased to inform you that your Article "Partner choice, confounding and trait convergence all contribute to phenotypic partner similarity", has now been accepted for publication in *Nature Human Behaviour*.

Please note that *Nature Human Behaviour* is a Transformative Journal (TJ). Authors whose manuscript was submitted on or after January 1st, 2021, may publish their research with us through the traditional subscription access route or make their paper immediately open access through payment of an article-processing charge (APC). Authors will not be required to make a final decision about access to their article until it has been accepted. IMPORTANT NOTE: Articles submitted before January 1st, 2021, are not eligible for Open Access publication. Find out more about Transformative Journals

With best regards,

Arunas Radzvilavicius, PhD
Editor, Nature Human Behaviour
Nature Research